# A GABAergic network from AVP- to VIP-neurons in the suprachiasmatic nucleus sets the timing of circadian behavior rhythms

Yubo Peng☯, Yusuke Tsuno ☯*, Takashi Maejima, Mohan Wang, Jaehun Jung, Ayako Matsui, Michihiro Mieda *

Department of Integrative Neurophysiology, Graduate School of Medical Sciences, Kanazawa University, Kanazawa, Japan

☯ These authors contributed equally to this work.
* tsuno@med.kanazawa-u.ac.jp (YT); mieda@med.kanazawa-u.ac.jp (MM)

## Abstract

The central circadian clock of the suprachiasmatic nucleus (SCN) consists of a network of multiple types of γ-aminobutyric acid (GABA)-ergic neurons and glial cells. However, the precise role of GABAergic transmission in the SCN remains unclear. In this study, we investigated the GABAergic regulation from arginine vasopressin (AVP)-producing neurons in the SCN shell to vasoactive intestinal polypeptide (VIP)-producing neurons in the SCN core. Blocking GABA release from AVP neurons via deletion of the vesicular GABA transporter (*Vgat*) gene lengthened the activity time (the interval between the onset and offset of locomotor activity) and shortened the duration of high $Ca^{2+}$ activity in VIP neurons to correspond to the behavioral rest time. Conversely, eliminating functional $GABA_A$ receptors ($GABA_AR$) in VIP neurons by in vivo genome editing reduced morning locomotor activity level and shortened the activity time, while lengthening the high $Ca^{2+}$ duration in VIP neurons. Optogenetic activation of AVP neurons in vivo increased $Ca^{2+}$ levels in VIP neurons during the night; this effect was significantly reduced in AVP neuron-specific *Vgat*-deficient mice. A similar $Ca^{2+}$ response in VIP neurons following AVP neuronal activation was observed in SCN slices and was inhibited by the $GABA_AR$ antagonist gabazine. Importantly, gabazine application alone elevated baseline $Ca^{2+}$ levels in VIP neurons, suggesting tonic GABA-mediated inhibition of these neurons. Moreover, AVP neuronal activation decreased $Ca^{2+}$ levels in non-AVP neurons located between AVP- and VIP-rich regions of the SCN. These results suggest that GABA released from AVP neurons indirectly disinhibits VIP neurons by suppressing intermediate non-AVP neurons, thereby precisely setting behavioral activity/rest time.

**Data availability statement:** All relevant data are within the paper and its Supporting information files.

**Funding:** This work was supported in part by JSPS KAKENHI Grant Numbers JP24KJ1189 (to Y.P.); JP23K06345 (to Y.T.); JP22K20738 (to A.M.); JP24K02137 (to T.M.); JP23K24064; JP25K02440, JP25K22589 (to M.M.) (Japan Society for the Promotion of Science: https://www.jsps.go.jp/english/index.html); JST SPRING Grant Number JPMJSP2135 (to Y.P., M.W., J.J) (Japan Science and Technology Agency: https://www.jst.go.jp/EN/); the Takeda Science Foundation (to M.M.) (https://www.takeda-sci.or.jp/en/); the Terumo Life Science Foundation (to M.M.) (https://www.terumo-zaidan.or.jp/english/); the Research Foundation for Opto-Science and Technology (to M.M.) (https://www.refost-hq.jp/); and the Koyanagi Foundation (to M.M.) (https://koyanagi-zaidan.com/). The funders had no role in study design, data collection and analysis, decision to publish, or preparation of the manuscript.

**Competing interests:** The authors have declared that no competing interests exist.

**Abbreviations:** ACSF, artificial CSF; AVP, arginine vasopressin; CNQX, 6-cyano-7-nitroquinoxaline-2,3-dione disodium; CT, circadian time; CV, coefficient of variation; D-AP5, D-(-)-2-amino-5-phosphonopentanoic acid; DD, constant darkness; GABA, γ-aminobutyric acid; GABA$_A$R, GABA$_A$ receptors; GPSCs, GABA$_A$R-mediated postsynaptic currents; PSCs, postsynaptic currents; SCN, suprachiasmatic nucleus; tTA, tetracycline transactivator; TTFL, transcription-translation feedback loop; Vgat, vesicular GABA transporter; VIP, vasoactive intestinal polypeptide.

## Introduction

The suprachiasmatic nucleus (SCN) of the hypothalamus serves as the central circadian clock in mammals, orchestrating multiple circadian biological rhythms in the body [1]. The SCN is composed of approximately 20,000 cells. Individual SCN cells can generate cellular circadian oscillations driven by the autoregulatory transcription-translation feedback loop (TTFL) of clock genes, including *Bmal1*, *Clock*, *Per1/2/3*, and *Cry1/2*. This TTFL-based molecular clock is a common property of most cells throughout the body. Crucially, within the SCN, intercellular communication, in addition to cell-autonomous TTFL oscillations, is essential for generating a robust and coherent circadian rhythm [1,2].

The SCN is a network of neurochemically heterogeneous γ-aminobutyric acid (GABA)-ergic neurons. Several types of GABA neurons can be distinguished by the co-expressed neuropeptides [1,2]. Arginine vasopressin (AVP)-producing GABAergic neurons in the shell (dorsomedial region) and vasoactive intestinal polypeptide (VIP)-producing GABAergic neurons in the core (ventrolateral region) represent two of the major neuron types in the SCN. VIP has been reported as a critical factor in the maintenance and synchronization of SCN neurons [3–6]. In addition, these neurons also contribute to the output pathway from the SCN [7–9]. In contrast, AVP neurons have been suggested to be the primary pacesetter cells that determine the period length of the circadian rhythm generated by the SCN network [10–13].

The functional roles of GABA-mediated signaling in the SCN network remain controversial [14]. GABA was initially reported to synchronize action potential firing rhythms of dispersed SCN neurons through GABA$_A$ receptors (GABA$_A$R) [15]. Studies using SCN slices and GABA$_A$R antagonists showed that GABA synchronizes or desynchronizes the cellular circadian oscillations depending on the lighting conditions before the slice preparation [16–18]. In addition, the regulation of extracellular GABA in the SCN by astrocytes was reported to be critical for circadian timekeeping in neonatal SCN cultures [19–21]. The SCN-specific in vivo deletion of the vesicular GABA transporter gene (*Vgat*, also called *Slc32a1*), which is necessary for filling synaptic vesicles with GABA and thus for synaptic GABA release [22], attenuated the circadian behavior rhythm, but the TTFL oscillation remained more or less normal [23,24].

We previously demonstrated that AVP neuron-specific deletion of *Vgat* drastically alters the daily locomotor activity pattern without changing the free-running period [25]. Namely, these mice showed a marked lengthening of the activity time in the circadian behavior rhythm due to an extended interval between evening and morning locomotor activities. GABA released by AVP neurons was suggested not to significantly affect the synchrony of TTFLs among SCN neurons but to regulate the phase relationships between TTFLs in the SCN and circadian morning/evening locomotor activities [25]. In addition, the daily rhythm of the in vivo multiunit activity (MUA) in the SCN clearly changed to an aberrant bimodal pattern that correlated with dissociated morning/evening locomotor activities [25]. These observations indicated that GABAergic transmission from AVP neurons regulates the activities of other SCN neurons to temporally restrict circadian behavior to appropriate time windows in SCN TTFLs [25]. In this study, we show the critical role of indirect GABAergic regulation of AVP neurons on VIP neurons in the activity/rest time setting.

## Results

### In vivo Ca²⁺ rhythm of SCN VIP neurons tracks the compressed rest period of the behavior rhythm in AVP neuron-specific *Vgat* deficiency

AVP neuron-specific *Vgat*-deficient mice (*Avp-Vgat⁻ᐟ⁻*) showed a lengthened activity time (the interval between the locomotor activity onset and offset) of behavior rhythm with little change in the intracellular Ca²⁺ rhythm of AVP neurons, resulting in a misalignment of locomotor activity and AVP neuronal Ca²⁺ rhythm [25]. To elucidate how GABA released from AVP neurons regulates the activity of VIP neurons in the SCN, we first examined the temporal relationship between the circadian behavioral rhythm and the intracellular Ca²⁺ in VIP neurons (VIP-Ca²⁺) of *Avp-Vgat⁻ᐟ⁻* mice in vivo using fiber photometry.

To specifically target the jGCaMP7s expression in VIP neurons of *Avp-Vgat⁻ᐟ⁻* mice, we introduced a *Vip-tTA* allele [26] into *Avp-Cre; Vgat^flox/flox* (*Avp-Vgat⁻ᐟ⁻*) or *Avp-Cre; Vgat^wt/flox* (control) mice. In *Vip-tTA* mice, the tetracycline transactivator (tTA) is expressed specifically in VIP neurons. We then specifically expressed the fluorescent Ca²⁺ indicator jGCaMP7s [27] in SCN VIP neurons by focally injecting a tTA-dependent AAV vector (AAV-*TRE-jGCaMP7s*) and implanted an optical fiber just above the SCN (Fig 1A and 1B) [12].

The VIP-Ca²⁺ rhythm was synchronized in antiphase with the locomotor activity rhythm in control (*Avp-Cre; Vgat^wt/flox; Vip-tTA*) mice (Figs 1C–1E, S1, and S2G) and showed high Ca²⁺ levels across the entire span of the behavioral rest period, as reported previously [12,28–30]. To quantitatively characterize the relationship between locomotor activity and VIP-Ca²⁺ rhythms, we set thresholds to define the onset and offset of the behavioral active phase and the high VIP-Ca²⁺ phase in the daily profiles of individual mice. For locomotor activity, the median daily locomotor activity level was regarded as the threshold (S1 Fig). For VIP-Ca²⁺, we examined three thresholds, i.e., 20%, 50%, and 80% of the peak-to-trough amplitude, because the shape of VIP-Ca²⁺ daily profiles was often skewed, especially in *Avp-Vgat⁻ᐟ⁻* mice.

In *Avp-Vgat⁻ᐟ⁻* mice (*Avp-Cre; Vgat^flox/flox; Vip-tTA*), the shape of the VIP-Ca²⁺ profile was altered, which correlated with the lengthening of behavioral activity time (onset-offset interval), as reported previously (Fig 1F and 1G) [25]. In LD, VIP-Ca²⁺ levels dropped earlier in the evening and began to rise earlier in the dawn (Figs 1F left and S2F left). In addition, VIP-Ca²⁺ took a longer time to increase and reach the plateau in both LD and DD (Fig 1J). Especially in DD, the circadian time (CT) at which VIP-Ca²⁺ rose to 50% or 80% of the maximum was significantly delayed, consistent with the later locomotor activity offset in *Avp-Vgat⁻ᐟ⁻* mice (Figs 1E–1G right, and S2F right). In contrast, VIP-Ca²⁺ offset phases did not differ between genotypes in DD, although this might be due to CT12 being defined by the locomotor activity onset in individual mice (Figs 1F right and S2F right). Consequently, the duration of high VIP-Ca²⁺, when thresholds were set at 50% or 80% of the maximum, was significantly shortened (50%: Control, 11.87 ± 0.11 h; *Avp-Vgat⁻ᐟ⁻*, 9.52 ± 0.67 h, $P = 0.002$; 80%: Control, 9.67 ± 0.30 h; *Avp-Vgat⁻ᐟ⁻*, 5.71 ± 0.39 h, $P < 0.0001$), complementary to the expansion of behavioral activity time (Control, 13.66 ± 0.50 h; *Avp-Vgat⁻ᐟ⁻*, 18.31 ± 0.26 h, $P < 0.0001$) in constant darkness (DD) (Fig 1H and 1I). Intriguingly, relative VIP-Ca²⁺ levels at locomotor activity onset and offset were similar between the two genotypes, indicating that the phase relationship between the VIP-Ca²⁺ rhythm and locomotor activity was essentially maintained in *Avp-Vgat⁻ᐟ⁻* mice (Figs 1D, 1E, and S2B). These results suggest that GABA released from AVP neurons regulates the timing of high VIP-Ca²⁺. The delayed duration of high VIP-Ca²⁺ caused by VGAT deficiency in AVP neurons may allow the morning locomotor activity to occur later. In LD, light is likely to directly drive VIP neuronal activity in the absence of AVP neuronal GABA release, resulting in a less severe phenotype compared with that observed in DD.

### Elimination of GABA_AR in SCN VIP neurons reduces morning locomotor activity and shortens the activity time of the behavior rhythm

Thus, SCN VIP neurons likely receive direct or indirect GABAergic regulation from AVP neurons, as well as GABA from other types of SCN neurons. To investigate the role of GABAergic signaling in VIP neurons in generating the behavior rhythm, we next aimed to eliminate ionotropic GABA_AR specifically in these neurons. GABA_AR is essentially a pentameric

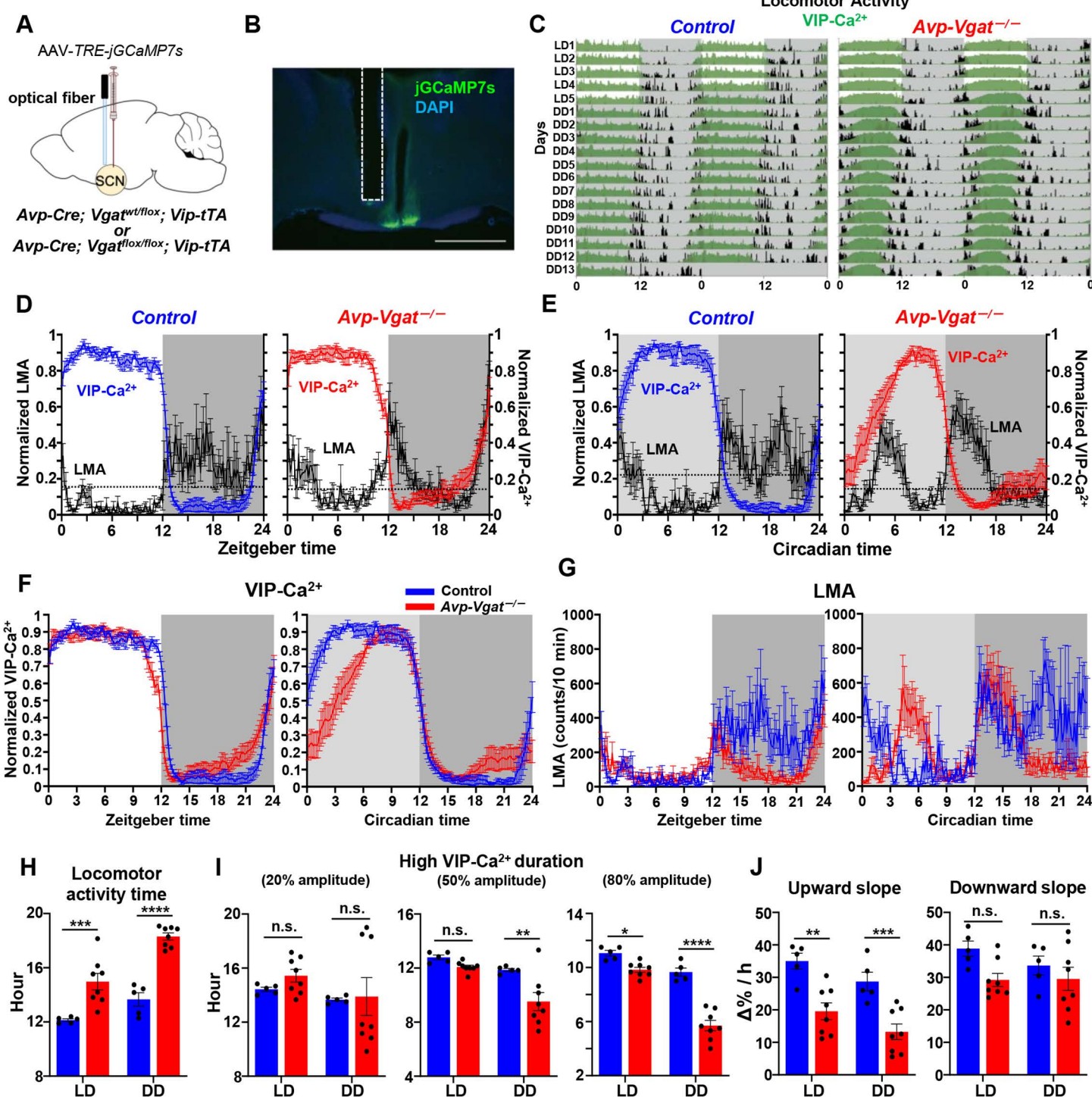

**Fig 1. VIP-Ca²⁺ rhythm corresponds to the compressed rest time of the locomotor activity in *Avp-Vgat⁻/⁻* mice. (A)** Schematic diagram of viral vector (*AAV-TRE-jGCaMP7s*) injection and optical fiber implantation at the SCN in control (*Avp-Cre; Vgat^wt/flox; Vip-tTA*) or *Avp-Vgat⁻/⁻* (*Avp-Cre; Vgat^flox/flox; Vip-tTA*) mice for fiber photometry recording. **(B)** A representative coronal section of mice with jGCaMP7s expression in SCN VIP neurons. A white dotted square shows the estimated position of the implanted optical fiber. Green, jGCaMP7s; blue, DAPI. Scale bar, 1 mm. **(C)** Representative plots of the in vivo jGCaMP7s signal of SCN VIP neurons (green: VIP-Ca²⁺) overlaid with locomotor activity (black: home-cage activity) in actograms. Detrended Ca²⁺ signals are shown with normalization in each row. Control (Left) and *Avp-Vgat⁻/⁻* (Right) mice were initially housed in LD (LD1 to LD5), then in DD

(DD1 to DD13). Gray shading indicates the time when the lights were off. **(D, E)** Population-averaged, normalized daily VIP-Ca²⁺ activity profiles (blue or red) overlaid with population-averaged, normalized locomotor activity (LMA) profiles (black) for control and *Avp-Vgat⁻ᐟ⁻* mice in LD (LD2-5, D) or in DD (DD8-12, E). The black dashed line indicates the median of averaged locomotor activity. In DD, CT12 was determined as the onset of locomotor activity (see S1 Fig for individual data). **(F)** Population-averaged normalized daily VIP-Ca²⁺ activity profiles for control (blue) and *Avp-Vgat⁻ᐟ⁻* (red) mice in LD (left) or in DD (right). The same dataset shown in Fig 1D and 1E is replotted to compare genotypes across lighting conditions (Left: LD; Right: DD). **(G)** Population-averaged daily locomotor activity profiles for control (blue) and *Avp-Vgat⁻ᐟ⁻* (red) mice in LD (left) or DD (right). The same dataset shown in Fig 1D and 1E is replotted without normalization to compare genotypes across lighting conditions (Left: LD; Right: DD). **(H)** Mean locomotor activity time (the interval between the onset and offset of locomotor activity) in LD or in DD. **(I)** Mean duration of high VIP-Ca²⁺ in LD (LD2-5) or in DD (DD8-12) calculated as the time above thresholds corresponding to 20% (left), 50% (middle), or 80% (right) of the peak-to-trough amplitude. **(J)** Mean speed of VIP-Ca²⁺ increase (left) and decline (right) in LD or in DD, which are expressed as slopes calculated by dividing the magnitude of VIP-Ca²⁺ change (60% change: from 20% to 80% or from 80% to 20%) by the time (h) required for that change. Blue, control (*Avp-Vgat⁺ᐟ⁻*, i.e., *Avp-Cre; Vgat ^wt/flox*, n = 5); Red, *Avp-Vgat⁻ᐟ⁻* (n = 8). Values are mean ± SEM. **P < 0.01; ***P < 0.001; ****P < 0.0001 by two-way mixed-design ANOVA post-hoc two-tailed Student *t* test with Bonferroni correction. The data underlying this figure can be found in S1 Data.

protein composed of α, β, and γ or δ subunits [31]. Each subunit has multiple subtype genes, but we had no information about which genes to delete. Considering that the β subunit is necessary for functional GABA_ARs and that only three subtype genes encode the β subunit (*Gabrb1–3*), we introduced indel mutations in all three simultaneously and specifically in VIP neurons by in vivo genome editing (*Vip-GABA_AR⁻ᐟ⁻* mice). We first crossed Cre-dependent SpCas9-expressing (*Rosa-LSL-SpCas9-2A-EGFP*) mice [32] with VIP neuron-specific Cre driver (*Vip-ires-Cre*) mice [33]. Then we injected the mixture of three AAV vectors, each expressing gRNAs targeting one of the *Gabrb* genes (AAV-*U6-gGabrb1, 2, 3-EF1α-DIO-mCherry*), into the SCN of these mice (Figs 2A, 2C, S3, and S4). Indeed, GABA_AR-mediated postsynaptic currents (GPSCs) disappeared almost completely in VIP neurons of SCN slices prepared from *Vip-GABA_AR⁻ᐟ⁻* mice, confirming the effectiveness of this method (Fig 2D and 2E).

In LD, *Vip-GABA_AR⁻ᐟ⁻* mice showed a daily locomotor activity rhythm comparable to controls. In DD, however, the morning locomotor activity of *Vip-GABA_AR⁻ᐟ⁻* mice was significantly reduced with earlier and less obvious locomotor offset (Fig 2B, 2F, and 2G), resulting in a shortened activity time (Control, 13.64 ± 0.31 h; *Vip-GABA_AR⁻ᐟ⁻*, 12.16 ± 0.15 h, P = 0.0001) (Fig 2H). In contrast, their free-running period in DD did not alter significantly (S5B Fig). These results suggest that the disinhibition of VIP neurons due to the absence of GABA_ARs may suppress the locomotor activity in the morning.

### The high Ca²⁺ duration in VIP neurons is lengthened in *Vip-GABA_AR⁻ᐟ⁻* mice

In *Avp-Vgat⁻ᐟ⁻* mice, the high VIP-Ca²⁺ duration of the VIP-Ca²⁺ rhythm was shortened while the behavioral activity time was lengthened (Fig 1). Therefore, we next investigated how the VIP-Ca²⁺ rhythm alters in vivo in *Vip-GABA_AR⁻ᐟ⁻* mice (Fig 3A and 3B).

The VIP-Ca²⁺ rhythm was synchronized in antiphase with the locomotor activity rhythm in both control and *Vip-GABA_AR⁻ᐟ⁻* mice (Figs 3C–3E, S6, and S7H). In *Vip-GABA_AR⁻ᐟ⁻* mice, the morning locomotor activity was reduced and ended earlier in DD (Figs 3G and S7F), as described above (Fig 2). Correspondingly, the shape of the VIP-Ca²⁺ profile was altered. VIP-Ca²⁺ levels began to rise earlier in both LD and DD and took a longer time to increase and reach the plateau (Figs 3F, 3J, and S7G). The speed of VIP-Ca²⁺ decline in the evening was also slower in *Vip-GABA_AR⁻ᐟ⁻* than in control mice. Consequently, the duration of high VIP-Ca²⁺ was significantly extended in these mice when thresholds were set at 20% or 50% of the maximum in DD (20%: Control, 13.48 ± 0.20 h; *Vip-GABA_AR⁻ᐟ⁻*, 15.05 ± 0.49 h, P = 0.0008; 50%: Control, 11.52 ± 0.19 h; *Vip-GABA_AR⁻ᐟ⁻*, 12.44 ± 0.35 h, P = 0.015), consistent with the shortened behavioral activity time observed in these mice (*Vip-GABA_AR⁻ᐟ⁻*, 12.09 ± 0.23 h; Control, 13.83 ± 0.23 h, P < 0.0001) (Fig 3H and 3I). Importantly, relative VIP-Ca²⁺ levels at locomotor activity onset and offset were similar between the two genotypes, except that the activity onset occurred at lower VIP-Ca²⁺ levels in *Vip-GABA_AR⁻ᐟ⁻* mice in LD, suggesting that the phase relationship between the VIP-Ca²⁺ rhythm and locomotor activity was essentially maintained in *Vip-GABA_AR⁻ᐟ⁻* mice (Figs 3D, 3E, and S7B). Thus, the earlier morning rise of VIP-Ca²⁺ may suppress the morning locomotor activity in *Vip-GABA_AR⁻ᐟ⁻* mice, underscoring the involvement of GABAergic input to VIP neurons in regulating VIP-Ca²⁺ and behavior rhythms.

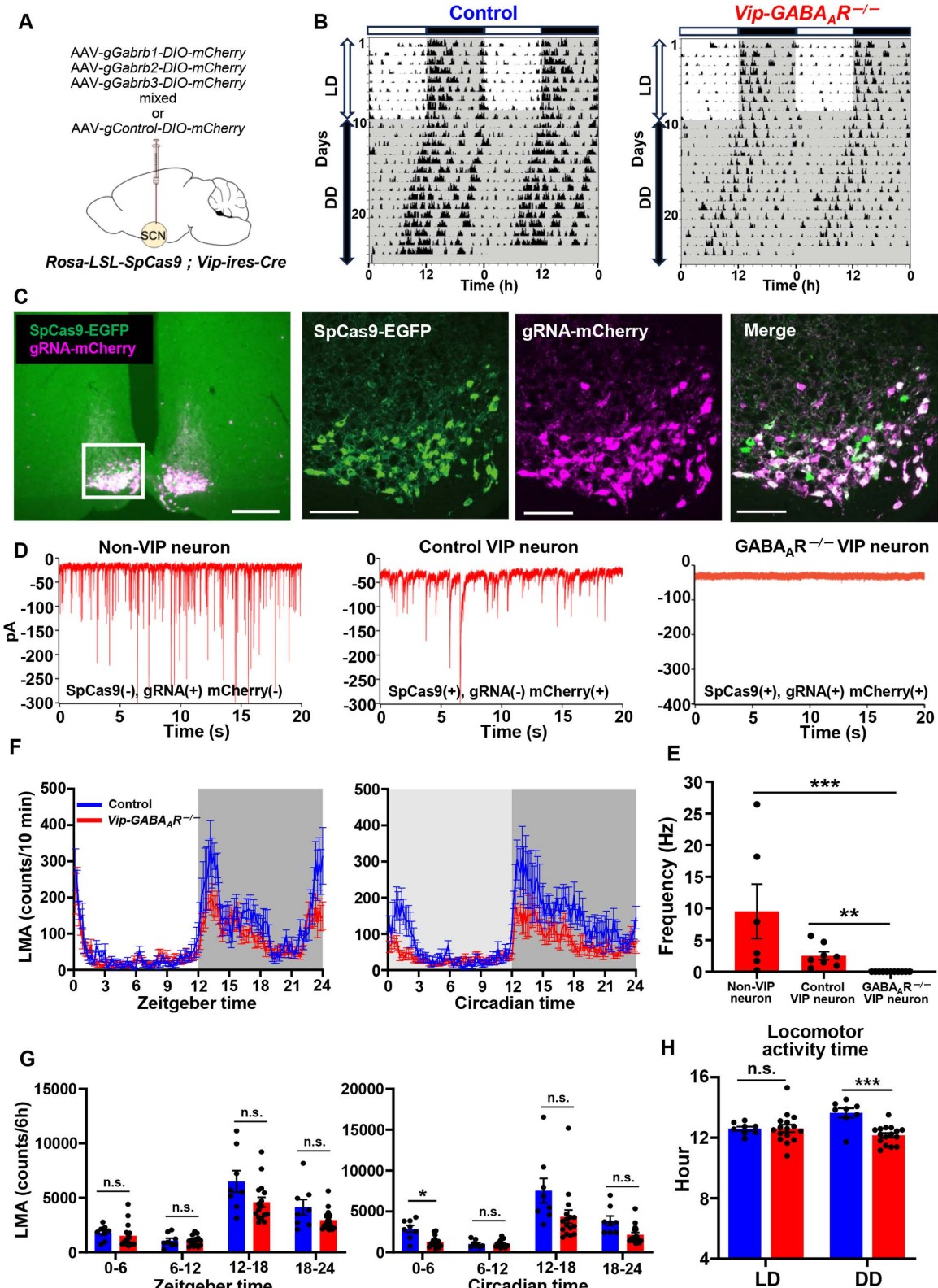

**Fig 2. *Vip-GABA_AR^{−/−}* mice reduce the locomotor activity and shorten the activity time. (A)** Schematic diagram of viral vector (AAV-*U6-gGabrb1,2,3-EF1α-DIO-mCherry* or AAV-*U6-gControl-EF1α-DIO-mCherry*) injection at the SCN in *Rosa26-CAG-LSL-SpCas9-2A-EGFP; Vip-ires-Cre*

mice to generate control or *Vip-GABA$_A$R$^{-/-}$* mice for locomotor activity recording. **(B)** Representative locomotor activity actograms of control and *Vip-GABA$_A$R$^{-/-}$* mice (home-cage activity). Gray shading indicates the time when the lights were off. **(C)** A representative coronal section of mice with SpCas9-2A-EGFP (green) and gRNA-DIO-mCherry (magenta) expression in SCN VIP neurons. The white rectangle indicates the region of the enlarged images (right). Scale bars, 200 μm (low magnification) or 50 μm (high magnification). **(D)** Representative spontaneous GPSCs (synaptic events) by whole-cell recordings from non-VIP neurons (SpCas9(-), gRNA(+) mCherry(-)), control VIP neurons (SpCas9(+), gRNA(-), mCherry(+)), and *GABA$_A$R$^{-/-}$* VIP neurons (SpCas9(+), gRNA(+) mCherry(+)). **(E)** Averaged spontaneous GPSC frequency of non-VIP neurons, control VIP neurons and *GABA$_A$R$^{-/-}$* VIP neurons. Since GPSCs had almost completely disappeared in *GABA$_A$R$^{-/-}$* VIP neurons, we assigned their frequency value to 0. **(F)** Averaged daily profile of locomotor activity (LMA) in LD (left) or DD (right). **(G)** Averaged locomotor activity counts per 6-hour interval in LD (left) or DD (right). **(H)** Mean activity time of locomotor activity rhythm in LD (left) or in DD (right). Blue, Control; red, *Vip-GABA$_A$R$^{-/-}$*. Values are mean ± SEM. $n = 6$ for non-VIP neurons, $n = 8$ for control VIP neurons, and $n = 11$ for *GABA$_A$R$^{-/-}$* VIP neurons **(D, E)**, $n = 8$ for control, $n = 16$ for *Vip-GABA$_A$R$^{-/-}$* mice **(F-H)**. *$P < 0.05$; **$P < 0.01$; ***$P < 0.001$ by Kruskal–Wallis test post-hoc Dunn's test **(E)**, or by two-way mixed-design ANOVA post-hoc two-tailed Student $t$ test with Bonferroni correction **(G, H)**. The data underlying this figure can be found in S1 Data.

Notably, VIP-Ca$^{2+}$ in *Vip-GABA$_A$R$^{-/-}$* mice showed greater fluctuations during the plateau phase, as indicated by an increased coefficient of variation (CV) (Figs 3F and S7C). The lack of GABA$_A$R might lead to an increase in basal Ca$^{2+}$ levels in VIP neurons. However, the current methodology does not allow determination of absolute Ca$^{2+}$ concentrations and therefore cannot directly test this possibility.

### Optogenetic activation of AVP neurons increases VIP-Ca$^{2+}$ in a time- and GABA-dependent manner in vivo

Previous studies have reported that AVP neuronal fibers make sparse contacts onto VIP neurons and that VIP neurons respond to the optogenetic activation of AVP neurons by increasing Ca$^{2+}$ at around ZT22 in vivo [12,34]. Therefore, to further investigate the functional connectivity between AVP and VIP neurons, we next tested the time-of-day and GABA dependency of VIP-Ca$^2$ response to the optogenetic activation of AVP neurons in vivo.

To this end, ChrimsonR-mCherry, a red light-gated cation channel [35,36], and jGCaMP7s were expressed specifically in AVP and VIP neurons, respectively, by injecting AAV-*CAG-FLEX-ChrimsonR-mCherry* and AAV-*TRE-FLEXoff-jGCaMP7s* into the SCN of control *Avp-Cre; Vgat$^{wt/flox}$; Vip-tTA* mice (Fig 4A–4C). Interestingly, optogenetic stimulation of AVP neurons increased VIP-Ca$^{2+}$ levels only at night, when basal Ca$^{2+}$ levels are low, but not during the day, when basal Ca$^{2+}$ levels are high (Fig 4D–4H). This lack of effect during the day may reflect a ceiling effect, potentially because AVP and VIP neuronal activities are already high under daytime conditions. We confirmed that such VIP-Ca$^{2+}$ responses were not observed when mCherry alone was expressed instead of ChrimsonR-mCherry (S8C and S8D Fig). These results suggest that AVP neurons can activate VIP neurons in vivo.

To further examine the contribution of GABAergic transmission to the effects of AVP neuronal activation, we measured VIP-Ca$^{2+}$ responses to ChrimsonR-mediated stimulation of AVP neurons in *Avp-Vgat$^{-/-}$* (*Avp-Cre; Vgat$^{flox/flox}$; Vip-tTA*) mice. In these mice, the nighttime increase in VIP-Ca$^{2+}$ induced by ChrimsonR stimulation was significantly reduced compared with control mice (Fig 4D–4H). These results suggest that GABA released from AVP neurons is the primary mediator of AVP neuron-driven activation of VIP neurons.

### GABA$_A$R antagonist increases VIP-Ca$^{2+}$ and inhibits the response of VIP-Ca$^{2+}$ to the optogenetic activation of AVP neurons in SCN slices

To further understand how AVP neurons regulate VIP neurons in the SCN network, we next recorded the VIP-Ca$^{2+}$ response to the optogenetic activation of AVP neurons in coronal slices of the middle SCN along the rostro-caudal axis. ChrimsonR-mCherry and jGCaMP7s were expressed in AVP and VIP neurons of *Avp-Cre; Vip-tTA* mice, respectively, by injecting AAV vectors into the SCN (Fig 5A and 5B). Then, SCN slices were prepared, and an optical fiber was placed close to the ChrimsonR-mCherry-positive region in the SCN for optogenetic activation (Fig 5B).

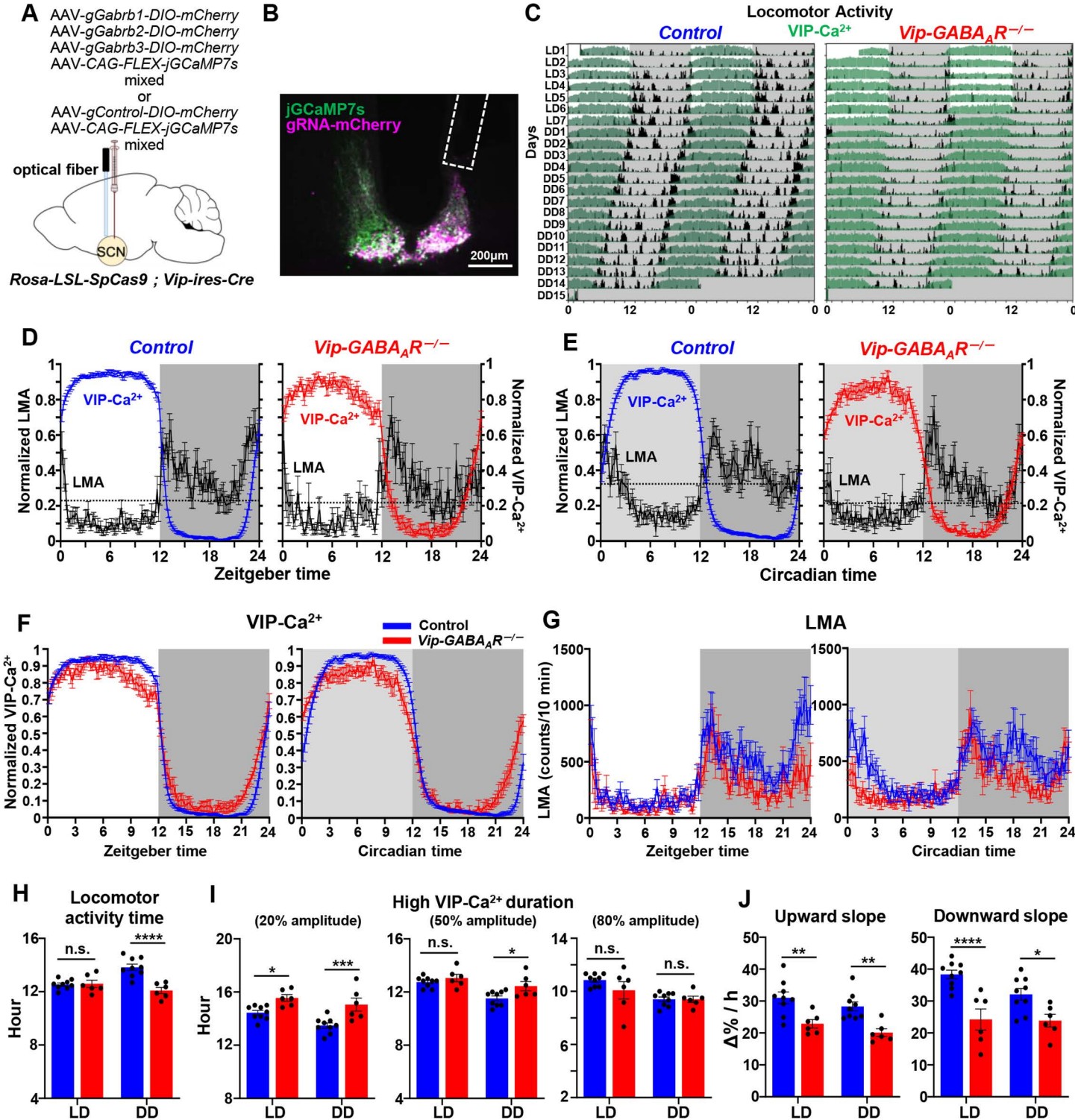

**Fig 3. The duration of high VIP-Ca²⁺ is prolonged in *Vip-GABA_AR⁻/⁻* mice.** **(A)** Schematic diagram of viral vector (AAV-*CAG-FLEX-jGCaMP7s* and AAV-*U6-gGabrb1,2,3-EF1α-DIO-mCherry* or AAV-*U6-gControl-EF1α-DIO-mCherry*) injection and optical fiber implantation at the SCN in *Rosa26-CAG-LSL-SpCas9-2A-EGFP; Vip-ires-Cre* mice to generate control or *Vip-GABA_AR⁻/⁻* mice for fiber photometry recording. **(B)** A representative coronal section of mice with jGCaMP7s expression and gRNA-DIO-mCherry expression in SCN VIP neurons. A white dotted square shows the estimated position of the

implanted optical fiber. Green, jGCaMP7s; magenta, mCherry. Scale bar, 200 μm. **(C)** Representative plots of the in vivo jGCaMP7s signal of SCN VIP neurons (green) overlaid with locomotor activity (black: home-cage activity) in actograms. Control (Left) and *Vip-GABA$_A$R$^{-/-}$* (Right) mice were initially housed in LD (LD1 to LD7) and then in DD (DD1 to DD15). Gray shading indicates the time when the lights were off. **(D, E)** Population-averaged, normalized daily VIP-Ca$^{2+}$ profiles (blue or red) overlaid with population-averaged, normalized locomotor activity (LMA) profiles (black) for control and *Vip-GABA$_A$R$^{-/-}$* mice in LD (LD3-7, D) or DD (DD8-14, **E**). The black dashed line indicates the median of averaged locomotor activity. In DD, CT12 was determined as the onset of locomotor activity (see S6 Fig for individual data). **(F)** Population-averaged, normalized daily VIP-Ca$^{2+}$ profiles for control (blue) and *Vip-GABA$_A$R$^{-/-}$* (red) mice in LD (left) or DD (right). The same dataset shown in Fig 3D and 3E is replotted to compare genotypes across lighting conditions (Left: LD; Right: DD). **(G)** Population-averaged daily locomotor activity profiles for control (blue) and *Vip-GABA$_A$R$^{-/-}$* (red) mice in LD (left) or DD (right). The same dataset shown in Fig 3D and 3E is replotted without normalization to compare genotypes across lighting conditions (Left: LD; Right: DD). **(H)** Mean locomotor activity time in LD or DD. **(I)** Mean duration of high VIP-Ca$^{2+}$ in LD (LD3-7) or in DD (DD8-14) calculated as the time above thresholds corresponding to 20% (left), 50% (middle), or 80% (right) of the peak-to-trough amplitude. **(J)** Mean speed of VIP-Ca$^{2+}$ increase (left) and decline (right) in LD or in DD, which are expressed as slopes calculated by dividing the magnitude of VIP-Ca$^{2+}$ change (60% change: from 20% to 80% or from 80% to 20%) by the time (h) required for that change. Blue, control ($n=9$); Red, *Vip-GABA$_A$R$^{-/-}$* ($n=6$). Values are mean±SEM. *$P<0.05$; **$P<0.01$; ***$P<0.001$; ****$P<0.0001$ by two-way mixed-design ANOVA post-hoc two-tailed Student $t$ test with Bonferroni correction. The data underlying this figure can be found in S1 Data.

Optogenetic activation of AVP neurons around ZT22 increased VIP-Ca$^{2+}$, as observed in vivo (Figs 5C, 5D, S9A, and S9B). Strikingly, the application of a GABA$_A$R antagonist, gabazine, alone raised the baseline VIP-Ca$^2$ (Figs 5E, 5F, and S9A), suggesting that VIP neurons are suppressed by GABA during the night. Moreover, the VIP-Ca$^{2+}$ increase induced by the optogenetic activation of AVP neurons was largely inhibited in the presence of gabazine (Figs 5G, 5H, S9A, and S9C). These data may be explained by a circuit in which GABA released from AVP neurons indirectly activates VIP neurons by inhibiting intermediate GABA neurons that suppress VIP neurons. On the other hand, the fact that gabazine failed to completely inhibit the VIP-Ca$^{2+}$ response may indicate the presence of parallel pathways mediated by AVP neuron-derived transmitters other than GABA, such as AVP and other neuropeptides. We also examined optogenetic stimulation around ZT14 and obtained results similar to those observed around ZT22 (S9D-S9G Fig).

### Optogenetic activation of SCN AVP neurons decreases Ca$^{2+}$ in the adjacent non-AVP cells

If AVP neurons disinhibit VIP neurons, as suggested in the previous section, there should be some population of intermediate GABA neurons that are inhibited by AVP neurons. To test this possibility, we made a spatial map of the Ca$^{2+}$ responses of non-AVP cells to the optogenetic activation of AVP neurons in SCN slices. To do this, we expressed ChrimsonR-mCherry and jGCaMP7s in AVP and non-AVP cells, respectively, by injecting AAV-*CAG-FLEX-ChrimsonR-mCherry* and AAV-*EF1-rDIO (reverse DIO)-jGCaMP7s* into the SCN of *Avp-Cre* mice (Fig 6A and 6B). Then, we prepared coronal slices of the middle SCN and monitored the jGCaMP7s signal throughout the slices.

The jGCaMP7s signals of non-AVP cells began to decrease shortly (~10 s) after the onset of optogenetic stimulation of AVP neurons in most areas of the SCN at around ZT22 (Figs 6C and S10A). As stimulation continued, the signals gradually recovered from this inhibitory response and eventually shifted to an increasing response in some areas including ventral region (Figs 6C–6E and S10). In contrast, cells in the intermediate region tended to exhibit stronger inhibitory responses with minimal excitation (Fig 6C–6E). Gabazine application alone increased non-AVP baseline Ca$^{2+}$ levels in both the middle and ventral regions (Fig 6F and 6G), suggesting that most SCN cells are suppressed by GABA during the night. Moreover, the non-AVP Ca$^2$ responses induced by the optogenetic activation of AVP neurons were largely abolished by gabazine application in both the middle and ventral regions (Fig 6H–6J). A certain degree of variability in spatiotemporal Ca$^{2+}$ response patterns between slices may derive from differences in the locations of coronal slice planes within the SCN, which are easily affected by slight difference in cutting angle and position and are thereby inevitable (S10A Fig). These results support the idea that AVP neurons indirectly activate VIP neurons via GABA by inhibiting another population of GABAergic neurons in the SCN network.

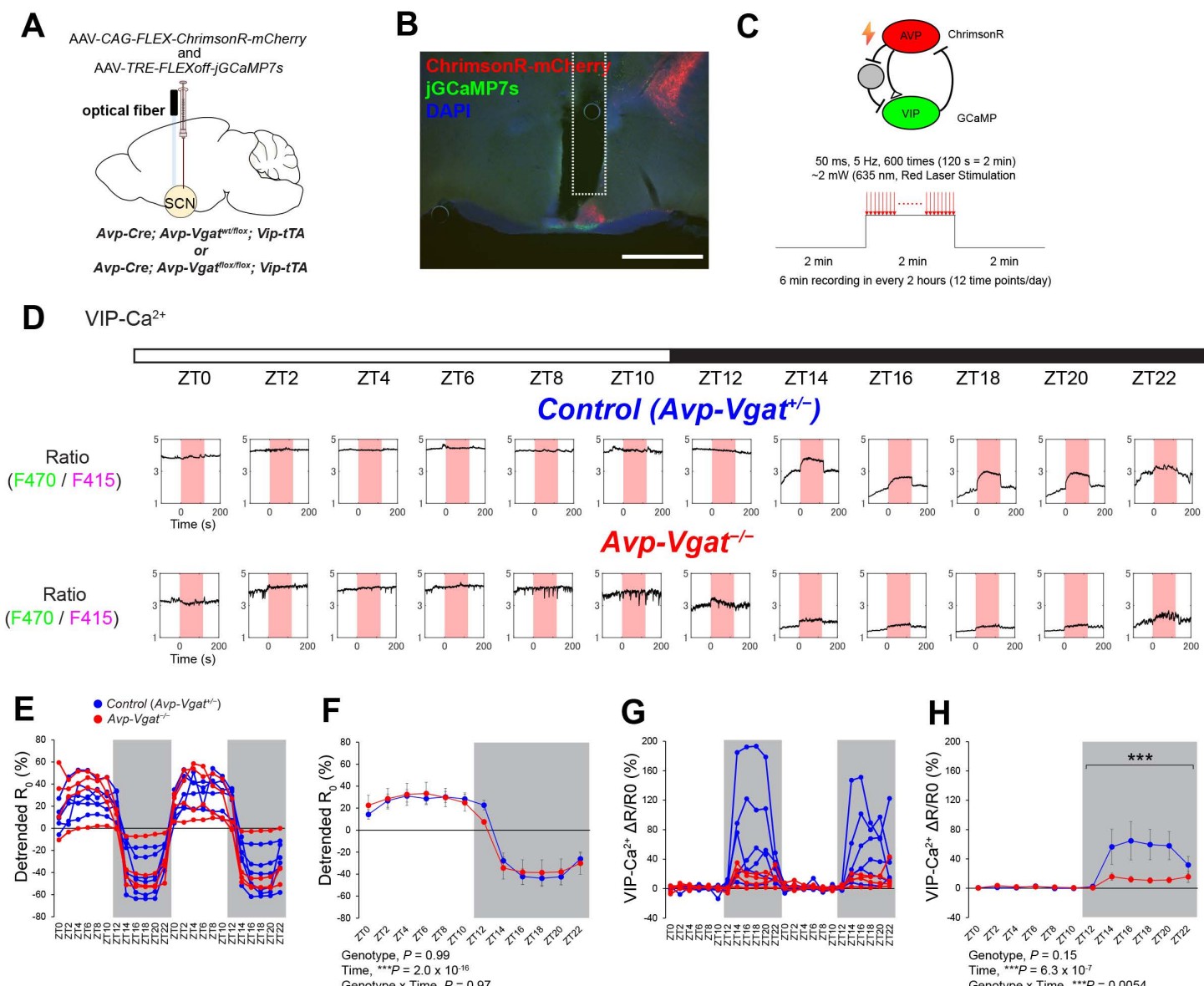

**Fig 4. VIP-Ca²⁺ response to optogenetic activation of AVP neurons decreases in *Avp-Vgat⁻/⁻* mice. (A)** Schematic diagram of viral vector (*AAV-CAG-FLEX-ChrimsonR-mCherry* and *AAV-TRE-FLEXoff-jGCaMP7s*) injection and optical fiber implantation at the SCN in control (*Avp-Cre; Avp-Vgat^wt/flox; Vip-tTA*) or *Avp-Vgat⁻/⁻* (*Avp-Cre; Avp-Vgat^flox/flox; Vip-tTA*) mice for fiber photometry recording of SCN VIP-Ca²⁺ and optogenetic stimulation of SCN AVP neurons. **(B)** A representative coronal section of mice with ChrimsonR-mCherry expression in SCN AVP neurons and jGCaMP7s expression in SCN VIP neurons. A white dotted square is estimated optical fiber position. Red, ChrimsonR-mCherry; Green, jGCaMP7s; Blue, DAPI. Scale bar, 1 mm. **(C)** Schematic diagram of recording and optogenetic stimulation protocol. Recording is performed 6 min in every 2 h. The recording includes pre-stimulation period (-120 to 0 s), during stimulation period (0−120 s), and post-stimulation period (120−240 s). **(D)** Representative traces of jGCaMP7s signal of SCN VIP neurons upon optogenetic stimulation of AVP neurons at 12 timing in vivo in Control (*Avp-Vgat⁺/⁻*, top) and *Avp-Vgat⁻/⁻* (bottom) mice. Ratio (R) calculated by F470 (Ca²⁺-dependent signal)/ F415 (Ca²⁺-independent control signal). Red shading indicates the timing of optical stimulation (635 nm, 50 ms pulse, 5 Hz, 120 s). **(E)** VIP-Ca²⁺ $R_0$ values were detrended using the 2-day average $R_0$ ($R_{48h}$) of each mouse as a reference and plotted. The detrended $R_0$ (%) was calculated as: ($R_0$ - $R_{48h}$)/ $R_{48h}$ x 100. **(F)** Population mean of detrended $R_0$ in (E). The detrended VIP-Ca²⁺ $R_0$ over two days was averaged by time of day to generate a 24-h profile per mouse. **(G)** VIP-Ca²⁺ $\Delta R/R_0$% in the late phase during the stimulation period (90−120 s) at 12 time points over two days. $\Delta R$ is the difference between the mean R-value of the late phase during the stimulation period ($R_1$, 90−120 s) and R0. **(H)** Population mean of VIP-Ca²⁺ $\Delta R/R_0$% in (G). (E−H) Blue, control (*Avp-Vgat⁺/⁻*, n=6); Red, *Avp-Vgat⁻/⁻* (n=4) mice. Error bars are SEM. White, day; dark gray, night. ***$P < 0.001$ by two-way mixed-design ANOVA with ZT as a within-subject (repeated-measures) factor and genotype as a between-subject factor, followed by two-tailed Welch's *t* test with Bonferroni correction. For Welch's *t* test, six time points were grouped into two time regions: day (ZT0−ZT10) and night (ZT12−ZT22). The data underlying this figure can be found in S1 Data.

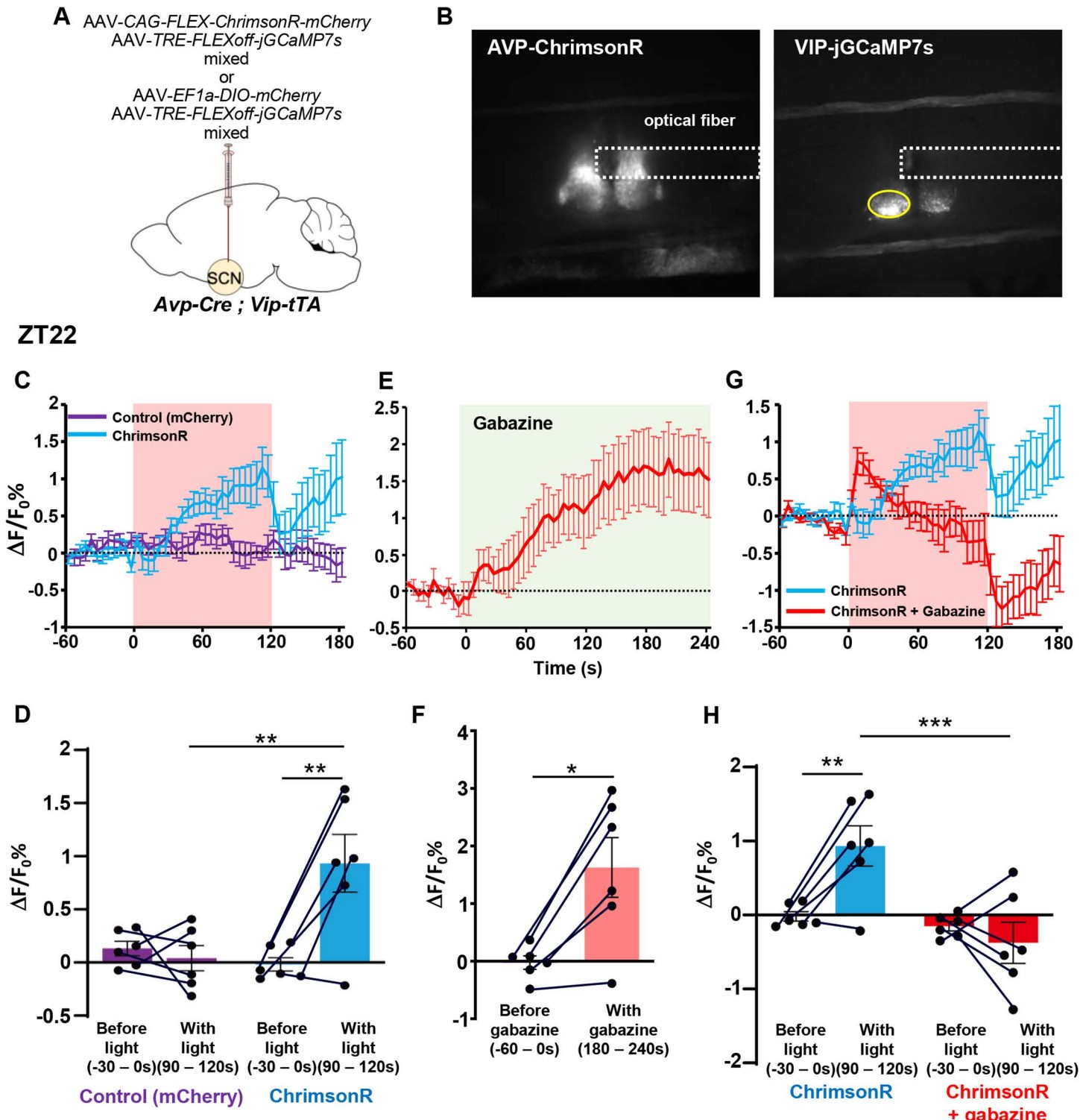

**Fig 5. GABA_AR antagonist increases basal VIP-Ca²⁺ and inhibits the response of VIP-Ca²⁺ to the optogenetic activation of AVP neurons in slices at ZT22.** **(A)** Schematic diagram of viral vector injection at the SCN in *Avp-Cre; Vip-tTA* mice for slice recording of VIP-Ca²⁺ and optogenetic stimulation of AVP neurons. **(B)** A representative coronal section of mice with ChrimsonR expression in AVP neurons (left) and jGCaMP7s expression in VIP neurons (right). White dotted squares indicate the estimated position of the optical fiber. The yellow circle indicates selected ROI for the subsequent

analyses. **(C)** Averaged traces of the jGCaMP7s signal of SCN VIP neurons aligned to optogenetic stimulation of AVP neurons at ZT22. ChrimsonR, blue; control (mCherry), purple. Red shading indicates the timing of optical stimulation (617 nm, 40 ms pulse, 10 Hz, 120 s). $F_0$ was defined as the mean F value during −60 to −30 s relative to stimulation onset. (see S9B Fig for individual data). **(D)** Mean VIP-Ca$^{2+}$ responses before (−30 to 0 s) and during the late phase of optogenetic stimulation (90–120 s in C). **(E)** Averaged trace of jGCaMP7s signal in VIP neurons aligned to bath application of gabazine (GABA$_A$R antagonist, 10 μM). The green shading indicates the period of gabazine application. $F_0$ was defined as the mean F value during −120 to −60 s relative to avoid overlap with comparison windows. **(F)** Mean VIP-Ca$^{2+}$ signals before (−60 to 0 s) and during the late phase of gabazine application (180–240 s) in (E). **(G)** Averaged trace of the jGCaMP7s signal of SCN VIP neurons aligned to optogenetic stimulation of AVP neurons in the presence of gabazine at ZT22 (red, ChrimsonR + Gabazine). For comparison, the trace recorded in the absence of gabazine (blue, ChrimsonR) is replotted from panel (C). Red shading indicates the timing of optical stimulation (617 nm, 40 ms pulse, 10 Hz, 120 s). $F_0$ was defined as the mean F value during −60 to −30 s relative to stimulation onset. (see S9C Fig for individual data). **(H)** Mean VIP-Ca$^{2+}$ responses to optogenetic stimulation of AVP neurons in the absence (blue) or presence (red) of gabazine, quantified before (−30 to 0 s) and during the late phase of optogenetic stimulation (90–120 s) in (G). Values are mean ± SEM. $n = 6$ for Control (mCherry) mice, $n = 6$ for ChrimsonR mice. $*P < 0.05$; $**P < 0.01$; $***P < 0.001$ by two-way mixed-design ANOVA post-hoc two-tailed Student $t$ test with Bonferroni correction or two-tailed Paired $t$ test. The data underlying this figure can be found in S1 Data.

## Discussion

In this study, we investigated the GABAergic network by which AVP neurons in the SCN shell regulate VIP neurons in the SCN core to set the timing of locomotor activity, highlighting the critical role of the GABAergic network in the SCN.

The *Vgat* deficiency in AVP neurons extended the locomotor activity time and compressed the duration of high VIP-Ca$^{2+}$ accordingly to fall within the behavioral rest time. Conversely, the GABA$_A$R deficiency in VIP neurons shortened the behavioral activity time by reducing the morning locomotor activity and lengthened the high VIP-Ca$^{2+}$ duration accordingly. Thus, GABAergic communication from AVP neurons to VIP neurons may contribute to the regulation of VIP-Ca$^{2+}$ dynamics within the SCN, which in turn is associated with the inverse relationship between locomotor activity time and high VIP-Ca$^{2+}$ duration. Moreover, the optogenetic activation of AVP neurons increased VIP-Ca$^{2+}$ levels both in vivo and ex vivo during the night, and this effect was inhibited by AVP neuron-specific VGAT deficiency in vivo and by GABA$_A$R antagonism in slices. Furthermore, baseline VIP-Ca$^{2+}$ levels increased upon GABA$_A$R antagonism, and AVP neuronal activation reduced Ca$^{2+}$ levels in cells located between AVP- and VIP-rich regions of the SCN. Together with our previous observation that the daily AVP-Ca$^{2+}$ rhythm begins to rise slightly before the onset of VIP-Ca$^{2+}$ [12], these findings support a model in which GABA released from AVP neurons indirectly activates VIP neurons by suppressing intermediate GABAergic neurons that inhibit VIP neurons. Activated VIP neurons then suppress the locomotor activity, primarily shaping the morning locomotor activity and subsequent rest timing, likely via downstream neurons regulating locomotion (Fig 7). Therefore, the lack of GABA release from AVP neurons may fail to disinhibit VIP neurons in the morning, leading to a delayed offset of the loco-motor activity. Conversely, loss of GABA$_A$R signaling in VIP neurons may reduce inhibition by intermediate GABA neurons in the morning, resulting in an earlier offset of the locomotor activity. It should be noted that not only neurons but also astrocytes may contribute as the intermediate GABA-releasing cells, as discussed later.

We do not exclude the possibility that altered AVP neuronal GABAergic control of extra-SCN areas may also contribute to the disturbance of behavioral profiles caused by VGAT deficiency in AVP neurons. However, as discussed previously [25], this possibility alone is unlikely to explain all of the observed phenomena. Attenuation of GABAergic output from SCN AVP neurons to the extra-SCN regions that regulate locomotor activity would be expected to affect only the magni-tude of locomotor activity, without altering its temporal pattern. However, this prediction is inconsistent with the phenotype observed in *Avp-Vgat$^{-/-}$* mice. In addition, following a change in lighting conditions from LD to DD, the daily profiles of locomotor activity and SCN VIP-Ca$^{2+}$ rhythms are gradually expanded and compressed, respectively, in these mice. This pattern may reflect a gradual shift in the coupling of multiple SCN oscillators, such as morning and evening oscillators, from a stable state observed in LD to that in DD [37]. Therefore, it is reasonable to conclude that impairment of the SCN central clock itself accounts, at least in part, for the phenotype of *Avp-Vgat$^{-/-}$* mice.

Importantly, *Avp-Vgat$^{-/-}$* and *Vip-GABA$_A$R$^{-/-}$* mice do not exhibit perfectly mirrored phenotypes. This observation may simply reflect the greater difficulty of compressing, rather than expanding, the locomotor activity time. Alternatively, it may

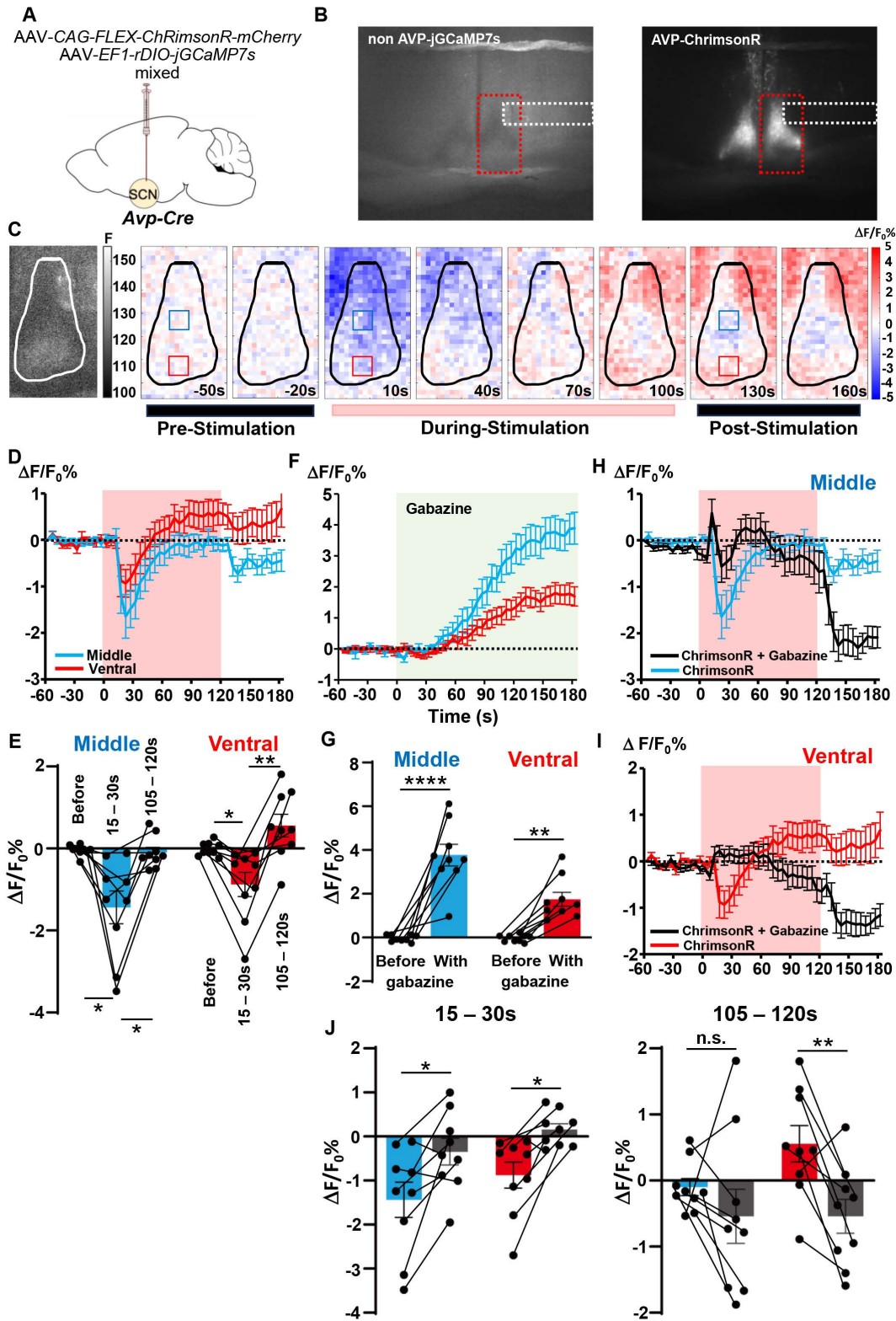

**Fig 6. Optogenetic activation of AVP neurons transiently decreases Ca²⁺ in the adjacent non-AVP cells. (A)** Schematic diagram of viral vector injection at the SCN in *Avp-Cre* mice for slice recording of SCN non-AVP cellular Ca²⁺ and optogenetic stimulation of AVP neurons. **(B)** A representative coronal section of mouse brain around SCN with jGCaMP7s expression in non-AVP cells (left) and ChrimsonR expression in AVP neurons (right).

White dotted squares indicate the estimated position of the optical fiber. The red rectangles indicate regions shown at higher magnification in panel **C.** (C) Left: Representative coronal section of *Avp-Cre* mice with jGCaMP7s expression in non-AVP cells (2.6 μm/pixel). The white outline indicates the regions considered as the SCN. Right: Representative pixel-level heat maps (20.8 μm/pixel) showing the jGCaMP7s signal from non-AVP cells in response to the optogenetic stimulation of AVP neurons at ZT22. Optical stimulation was applied from time 0 to 120 s. Blue and red squares on the maps indicate selected ROI (4 × 4 pixels) considered as the middle and ventral regions, respectively, for the subsequent analyses. **(D)** Averaged traces of the jGCaMP7s signals in the middle (blue) and ventral (red) regions aligned to optogenetic stimulation of AVP neurons. Red shading indicates the timing of optical stimulation (617 nm, 40 ms pulse, 10 Hz, 120 s) (see S10B Fig for individual data). **(E)** Mean jGCaMP7s signal in the middle (blue) and ventral (red) regions before (−15 to 0 s in D) and in the early (15–30 s) and late (105–120 s) stages during optogenetic stimulation. $F_0$ is defined as the mean fluorescence signal during −60 to −30 s. **(F)** Averaged traces of jGCaMP7s signal in the middle (blue) and ventral (red) regions aligned to bath application of gabazine (GABA$_A$R antagonist, 10 μM). The green shading indicates the period of gabazine application. **(G)** Mean jGCaMP7s signal in the middle (blue) and ventral (red) regions before (−30 to 0 s in F) and during the last 30 s of gabazine application (150–180 s in F). **(H, I)** Averaged traces of the jGCaMP7s signal in the middle (H) and ventral (I) regions aligned to optogenetic stimulation of AVP neurons at ZT22 without (ChrimsonR, blue or red) or with (ChrimsonR + Gabazine, black) gabazine application. Red shading indicates the timing of optical stimulation (617 nm, 40 ms pulse, 10 Hz, 120 s). The ChrimsonR traces in (D) and (H, I) are identical. **(J)** Mean jGCaMP7s signal in the middle (blue) and ventral (red) regions in the early (15–30 s in H or I, left) and late (105–120 s, right) stages during optogenetic stimulation in (H) and (I). Values are mean ± SEM. $n = 9$ for ChrimsonR mice. *$P < 0.05$; **$P < 0.01$; ***$P < 0.001$; ****$P < 0.0001$ by two-way mixed-design ANOVA post-hoc two-tailed Student $t$ test with Bonferroni correction. The data underlying this figure can be found in S1 Data.

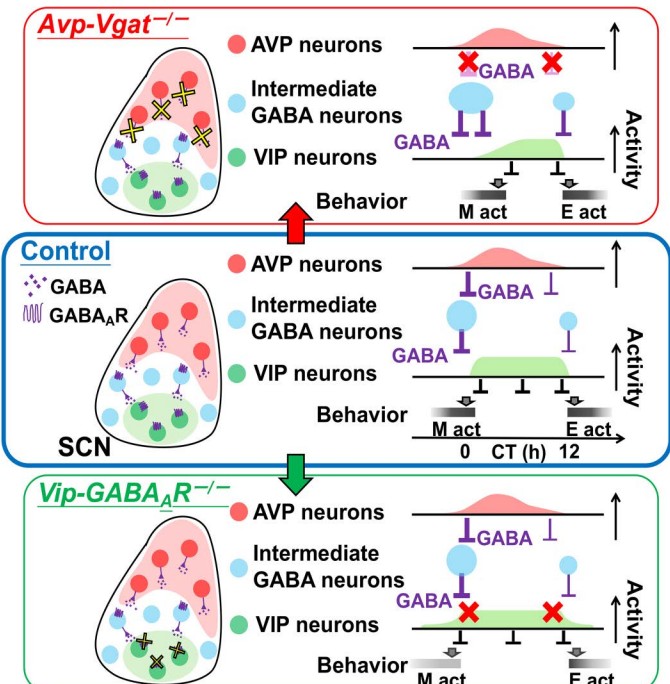

**Fig 7. A model of how the GABAergic network from AVP neurons to VIP neurons in the SCN sets the activity/rest time of the circadian behavior rhythm.** In normal mice (middle), GABA released from AVP neurons decreases the inhibitory effect of the intermediate GABA neurons on VIP neurons, particularly in the morning and to a lesser extent in the evening. This disinhibition enhances VIP neuronal activity and thereby sets the timing of morning (M) locomotor activity, while also contributing to the timing of evening (E) activity. In *Avp-Vgat$^{-/-}$* mice (top), the lack of GABA release from AVP neurons fails to disinhibit VIP neurons, especially in the morning, leading to a later offset of the locomotor activity. In *Vip-GABA$_A$R$^{-/-}$* mice (bottom), VIP neurons are insensitive to GABAergic inhibition by the intermediate GABA neurons, predominantly affecting morning regulation and resulting in an earlier offset of the locomotor activity. The actual daily activity pattern of the intermediate neurons remains unknown. Besides, astrocytes may contribute as the intermediate GABA-releasing cells. These points will be further discussed in the manuscript.

indicate that the signaling from AVP neurons to VIP neurons is unlikely to be mediated by a single linear pathway. AVP neurons are expected to regulate multiple targets in parallel, and VIP neurons likewise receive convergent inputs from sources beyond the AVP neuron–intermediate cell pathway described here. Nevertheless, the results of this study demonstrate that the GABAergic network from AVP neurons to VIP neurons plays a functionally significant role in determining the activity/rest time.

Both inhibitory and excitatory effects of GABA on SCN neurons have been reported, and these effects may vary depending on the time of the day, the region within the SCN, or photoperiod [14,17,38–40]. Therefore, we also considered the possibility that GABA released from AVP neurons directly excites VIP neurons. However, the observation that gabazine application increased VIP-$Ca^{2+}$ levels in SCN slices indicates that GABA inhibits VIP neurons as a population. This finding is consistent with a previous study reporting that GABA exerts a predominantly inhibitory effect in the ventral SCN [17]. In addition, AVP neurons send only sparse direct projections to VIP neurons [12,34]. In contrast, numerous GABA neurons located between AVP-rich and VIP-rich regions of the SCN receive dense AVP fiber inputs and project robustly to VIP neurons [34]. Indeed, we confirmed the presence of cells in these intermediate regions that respond to AVP neuronal activation with a reduction in $Ca^{2+}$ levels. Collectively, the above-mentioned disinhibitory pathway mediated by the intermediate neurons is likely the primary regulatory mechanism, although an additional direct GABAergic pathway may exist. In addition, not only fast synaptic, but also slower peri-synaptic and extra-synaptic GABAergic transmission may need to be considered. Furthermore, in addition to GABA, other transmitters, such as AVP, may play an additional role in linking AVP neurons to VIP neurons.

It remains to be determined whether the presumed intermediate neurons constitute a specialized cell type with a specific neurochemical identity or represent a heterogeneous population. GRP- or calretinin-expressing neurons may represent candidate intermediate neurons, as they are GABAergic neurons located between AVP and VIP neurons, receive contacts from AVP fibers, and project to VIP neurons [34]. On the other hand, because the SCN contains diverse populations of GABA neurons, the intermediate neurons may be more broadly distributed, both spatially and neurochemically. In terms of neuronal activity, one might initially expect the presumed intermediate neurons to be more active during the night, thereby inhibiting VIP neurons, and to be inhibited by AVP neurons. However, population-level MUA recordings have shown that SCN neurons are generally more active during the day, with a peak around midday [1], which appears inconsistent with this simple expectation. Importantly, this apparent contradiction may reflect the fact that the proposed disinhibitory circuit does not operate to define the peak or trough of VIP neuronal activity. Instead, firing phases across individual SCN neurons are broadly distributed, and SCN MUA spans the transitions at dawn and dusk [25,41,42]. We therefore propose that AVP neuron-derived GABA acts primarily during these transitional temporal windows to fine-tune the timing of locomotor activity offset and onset by transiently disinhibiting VIP neurons, rather than driving sustained changes in their overall activity level. Consistent with this view, VIP-$Ca^{2+}$ levels remained high at midday even in $Avp\text{-}Vgat^{-/-}$ mice, and low at midnight in $Vip\text{-}GABA_AR^{-/-}$ mice, indicating that peak and trough VIP neuronal activity are largely regulated by GABA-independent mechanisms, likely involving TTFL-driven cellular clocks. From this perspective, the presumed intermediate neurons need not necessarily be rhythmic for this model to hold.

Recent studies using organotypic SCN cultures have demonstrated the critical role of astrocytes in circadian timekeeping by regulating the extracellular GABA levels through rhythmic GABA uptake and release, as well as glutamate release that stimulates GABA release from nerve terminals [19–21]. Importantly, astrocytes drive daily fluctuations in extracellular GABA, with levels being higher at night. Therefore, it is tempting to hypothesize that GABA released from AVP neurons may modulates astrocyte activity through as-yet-unknown mechanisms, potentially reducing local GABA levels in the vicinity of VIP neurons. Alternatively, GABAergic neurons located in regions just outside the SCN, where neuronal activity rhythms may be in antiphase, could conceivably act as the presumed intermediate GABAergic neurons.

Although most SCN neurons contain GABA and express $GABA_AR$, they constitute heterogeneous populations and form asymmetric neural and humoral networks, indicating functional differentiation among cell types. In particular, GABAergic

communication between the SCN shell and core is likely to be far from symmetric [17,18,43]. Thus, it is essential to pay close attention to the directionality of GABAergic signaling when investigating the function of GABA in the SCN central clock. To this end, *Vgat* has previously been deleted to eliminate GABA release specifically from VIP or AVP neurons; deletion in VIP neurons caused little effect [24,44], whereas deletion in AVP neurons lengthened the activity time of the behavior rhythm [25]. However, until recently, there had been no way to inhibit GABAergic signaling in a cell type-specific manner without detailed knowledge of GABA$_A$R subtype genes expressed in the target cells. Here, we achieved complete cell type-specific blockade of GABA$_A$R signaling by deleting all three genes encoding the requisite β-subunits using in vivo genome editing. A similar strategy to simultaneously disrupt multiple subtype genes should be applicable to studying the cell type-specific functions of other ion channels with multiple subtypes, such as AMPA receptors and potassium channels. Moreover, it is noteworthy that our approach—recording the VIP-Ca$^{2+}$ responses to optogenetic stimulation of AVP neurons using a combination of Cre/tTA driver mice and Cre/tTA-dependent AAV vectors—enabled us to interrogate the directional GABAergic regulatory network from AVP neurons to VIP neurons.

Although the locomotor activity time was altered, the free-running period did not change in either *Avp-Vgat*$^{-/-}$ or *Vip-GABA$_A$R*$^{-/-}$ mice. Other mouse models with genetic impairments of VGAT or GABA$_A$R likewise exhibit free-running periods within the normal range, whereas the amplitudes of circadian rhythms are reduced to varying degrees [23,24,45]. We previously showed that clock gene expression rhythms are not significantly altered in either the shell or core of the SCN in *Avp-Vgat*$^{-/-}$ mice, suggesting that TTFLs tick normally in the absence of GABA release of AVP neurons [25]. Thus, AVP neuron-derived GABA may modify the temporal pattern of VIP neuronal activity with little effect on the TTFL. On the other hand, AVP neurons have been proposed to function as the primary pacesetter cells that determine the SCN ensemble period, a role that does not require GABA from these neurons [10–12,25,46]. Therefore, AVP neurons may use GABA to set on-off timers on their cellular clocks, thereby controlling the phase relationship between evening and morning locomotor activities. Taken together, AVP neurons may regulate both the pacesetting of the SCN ensemble rhythm and the phase-setting of evening and morning locomotor activities via partially independent transmitter systems.

## Materials and methods

### Ethics statements

All experimental procedures were approved by the Kanazawa University Animal Experiment Committee (approval numbers: AP-224337) and the Kanazawa University Safety Committee for genetic recombinant experiments (approval numbers: Kindai 6-2626, Kindai 6-2891). The study was carried out in compliance with Fundamental Guidelines for Proper Conduct of Animal Experiment and Related Activities in Academic Research Institutions under the jurisdiction of the Ministry of Education, Culture, Sports, Science and Technology (Notice No. 71 of 2006), and Standards relating to the Care and Keeping and Reducing Pain of Laboratory Animals (Notice of the Ministry of the Environment No. 88 of 2006).

### Animals

*Avp-Cre* BAC transgenic (C57BL/6J-*Tg(Avp-icre)#Meid*/Rbrc, RBRC12048) [11] and *Vip-tTA* knock-in (B6(Cg)-*Vip*$^{em1(tTA2)Miem}$/Rbrc, RBRC12109) [26] mice were reported previously. *Vip-ires-Cre* (*Vip*$^{tm1(cre)Zjh}$/J, JAX:010908) [33], *Vgat*$^{flox}$ (*Slc32a1*$^{tm1Lowl}$/J, JAX:012897) [47], and *Rosa26-LSL-SpCas9-2A-EGFP* mice (B6J.129(B6N)-*Gt(ROSA)26Sor*$^{tm1(CAG-cas9*,-EGFP)Fezh}$/J, JAX:026175) [32] were obtained from Jackson Laboratory. All lines were congenic on C57BL/6. We compared the conditional knockouts with controls whose genetic backgrounds were comparable. *Avp-Cre*, *Vip-ires-Cre*, *Vip-tTA*, and *Rosa26-LSL-SpCas9-2A-EGFP* mice were used in hemizygous or heterozygous condition. We used both male and female mice in our experiment. Whether we pooled data from both sexes or analyzed them separately, the conclusions we reached remained the same. Mice were maintained under a strict 12-h light/12-h dark cycle in a temperature- and humidity-controlled room and fed ad libitum.

## Viral vector and surgery

The AAV-2 ITR containing plasmids *pGP-AAV-CAG-FLEX-jGCaMP7s-WPRE* (Addgene plasmid #104495, a gift from Dr. Douglas Kim and GENIE Project) [27]. *pAAV-TRE-jGCaMP7s* was described previously [48]. In addition, we modified this plasmid to make an improved version by using an EcoRI-HindIII fragment of this plasmid containing *jGCaMP7s* sequence to replace an EcoRI-HindIII fragment containing *ChrimsonR-mCherry* from *pAAV-TRE-ChrimsonR-mCherry* (Addgene #92207, a gift from Alice Ting) [36]. *pAAV-U6-gGabrb1~3-EF1α-DIO-mCherry*, plasmids for CRISPR-Cas9-mediated *Gabrb1~3* gene disruption, were generated as follows. The target sites for CRISPR-Cas9 were designed by CRISPOR (http://crispor.tefor.net/) [49]. Two sequences targeting each *Gabrb* gene were selected: *Gabrb1*, 5′-AAGGATATGACATTCGCTTG-3′ and 5′-CGCATCCCGACGTCCACCGG-3′; *Gabrb2*, 5′- TGACCCTAGTAATATGTCGC-3′ and 5′-ATGTTCATTCCTACGGCCAC-3′; *Gabrb3*, 5′- ATTCGCCTGAGACCCGACTT-3′ and 5′-CGACATCGCCAGCATC GACA-3′. Oligonucleotides encoding the guide sequences were cloned into the BbsI and BsaI sites of *pX333* (Addgene #64073, a gift from Dr. Andrea Ventura) [50]. Then, a fragment containing two tandem units of *U6-gRNA* was amplified by PCR, using the following primers: 5′-agtacgcgTCGAGCATGCTCGAGAATGG-3′ and 5′-agtacgcgtCGGGTA CCCCATTTGTCTGC-3′, and cloned into the MluI site of *pAAV-EF1a-DIO-mCherry* (a gift from Dr. Bryan Roth) as described previously [48]. *pAAV-U6-gGabrb2-EF1α-DIO-mCherry* happened to contain two copies of the amplified fragment. *pAAV-U6-gControl-EF1α-DIO-mCherry* contains spacer sequences from *pX333* instead of gRNA sequences for *Gabrb* genes. *pAAV-CAG-FLEX-ChrimsonR-mCherry* was described previously [12]. As a negative control for the optogenetic study, we injected AAV-*EF1α-DIO-mCherry* or AAV-*EF1α-DIO-hM3Dq-mCherry*, which was generated with plasmids *pAAV-EF1α-DIO-mCherry* or *pAAV-EF1α-DIO-hM3Dq-mCherry* provided by Dr. Bryan Roth, University of North Carolina [51]. *pAAV-TRE-FLEXoff-jGCaMP7s* was made by replacing an EcoRI-SpeI fragment containing *ChrimsonR-mCherry* of *pAAV-TRE-ChrimsonR-mCherry* with an EcoRI-XbaI fragment containing *jGCaMP7s* from *pGP-AAV-CAG-FLEX-jGCaMP7s-WPRE*. *pAAV-EF1-rDIO-jGCaMP7s* was made by replacing an AscI-NheI fragment containing *ChR2-EYFP* of *pAAV-DIO-hChR2(H134R)-EYFP-WPRE-pA* (provided by Dr. Karl Deisseroth, Stanford University) with a *jGCaMP7s* cDNA fragment amplified by PCR from *pGP-AAV-CAG-FLEX-jGCaMP7s-WPRE,* using the following primers: 5′-ataggcgcGCCACCATGGGTTCTCATCA-3′ and 5′-gcgactagTCACTTCGCTGTCATCATTTG-3′. Note that gene expression from AAV-*TRE-FLEXoff-jGCaMP7s* and AAV-*EF1-rDIO-jGCaMP7s* stops upon Cre-mediated recombination. Recombinant AAV vectors (AAV2-rh10) were produced using a triple-transfection, helper-free method and purified as described previously [11]. The titers of recombinant AAV vectors were determined by quantitative PCR: AAV-*CAG-DIO-jGCaMP7s*, $3.4 \times 10^{13}$; AAV-*TRE-jGCaMP7s*, $6.3 \times 10^{11}$; AAV-*TRE-jGCaMP7s* (improved), $5.8 \times 10^{12}$; AAV-*TRE-FLEXoff-jGCaMP7s*, $1.48 \times 10^{13}$; AAV-*EF1-rDIO-jGCaMP7s*, $1.5 \times 10^{13}$; AAV-*U6-gGabrb1-EF1α-DIO-mCherry*, $6.06 \times 10^{12}$; AAV-*U6-gGabrb2-EF1α-DIO-mCherry*, $7.42 \times 10^{12}$; AAV-*U6-gGabrb3-EF1α-DIO-mCherry*, $6.56 \times 10^{12}$; AAV-*U6-gControl-EF1α-DIO-mCherry*, $2.6 \times 10^{12}$; AAV-*EF1α-DIO-hM3Dq-mCherry*, $4.5 \times 10^{12}$; AAV-*CAG-FLEX-ChrimsonR-mCherry*, $1.5 \times 10^{13}$; and AAV-*EF1α-DIO-mCherry*, $5.2 \times 10^{12}$ genome copies/ml. Stereotaxic injection of AAV vectors was performed as described previously [10]. Two weeks after surgery, we began monitoring the mice for their locomotor activity.

## In vivo fiber photometry

We used 8 *Avp-Vgat$^{-/-}$ × Vip-tTA* (*Avp-Cre; Vgat$^{flox/flox}$; Vip$^{wt/tTA}$*) mice, 5 control mice (*Avp-Cre; Vgat$^{wt/flox}$; Vip$^{wt/tTA}$*), and 15 *Vip-ires-Cre; Rosa26-LSL-SpCas9-2A-EGFP* mice for the GABA$_A$R disruption study (6 or 9 for *gGabrb1-3* or *gControl*). The mice were anesthetized by administering a cocktail of medetomidine (0.3 mg/kg), midazolam (4 mg/kg), and butorphanol (5 mg/kg) and were secured at the stereotaxic apparatus (Muromachi Kikai). Lidocaine (1%) was applied for local anesthesia before making the surgical incision. We drilled small hole in the exposed region of the skull using a dental drill. We injected 0.5–1.0 µL of the virus (AAV-*U6-*gGabrb1*-3-EF1α-DIO-mCherry*; AAV-*CAG-FLEX-jGCaMP7s* mixed (*gGabrb1: gGabrb2: gGabrb3: jGCaMP7s* = 1:1:1:2) or AAV-*U6-gControl-EF1α-DIO-mCherry*; AAV-*CAG-FLEX-jGCaMP7s*

mixed or *AAV-TRE-jGCaMP7s*) (flow rate = 0.1 μL/min) at the right or bilateral of SCN (posterior: 0.5 mm, lateral: 0.25 mm, depth: 5.7 mm from the bregma) with a 33 G Hamilton Syringe (1701RN Neuros Syringe, Hamilton) to label VIP neurons. Subsequently, we placed an implantable optical fiber (400 μm core, N.A. 0.39, 6 mm, ferrule 2.5 mm, FT400EMT-CANNULA, Thorlabs or RWD) above the SCN (posterior: 0.2 mm, lateral: 0.2 mm, depth: 5.2–5.4 mm from the bregma) with self-adhesive resin cement (Super-bond C&B, Sun Medical or RelyX Unicem2 Automix, 3M ESPE). The cement was painted black. Atipamezole (0.3 mg/kg) was administered postoperatively to reduce the anesthetized period. The mice were used for experiments 2–8 weeks after the virus injection and optical fiber implantation. Their ages ranged from 3 to 14 months old, including both males and females.

A single-color fiber photometry system (COME2-FTR, Lucir) was used to record the Ca$^{2+}$ signal of SCN neurons in freely moving *Avp-Vgat*$^{-/-}$ mice (Figs 1 and S1) [12,52]. Fiber-coupled LED (M470F3, Thorlabs) with LED Driver (LEDD1B, Thorlabs) was used as an excitation blue light source. The light was reflected by a dichroic mirror (495 nm), went through an excitation bandpass filter (472/30 nm), and then to the animal via a custom-made patch cord (400 μm core, N.A. 0.39, ferrule 2.5 mm, length 50 cm, COME2-FTR/MF-F400, Lucir) and the implanted optical fiber. We detected the jGCaMP7s fluorescence signal by a photomultiplier through the same optical fibers and an emission bandpass filter (520/36 nm); furthermore, we recorded the signal using Power Lab (AD Instruments) with Lab Chart 8 software (AD Instruments). The excitation blue light intensity was 10–30 μW at the tip of the patch cord of the animal side. We recorded the same for 30 s every 10 min for 2 weeks to reduce photobleaching. During the recording, the mouse was housed in a 12-h light-dark cycle for more than 5 days (LD condition) and then moved to continuous darkness for ~10 days (DD condition) in a custom-made acrylic cage surrounded by a sound-attenuating chamber. A rotary joint for the patch cord was stopped during the recording to prevent artificial baseline fluctuation. The animal's locomotor activity was monitored using an infra-red sensor (Supermex PAT.P and CompACT AMS Ver. 3, Muromachi Kikai). The detected GCaMP signal was averaged within a 30-s session [25].

Another dual-color fiber photometry system (FP3002, Neurophotometrics) was used to record the calcium signal of SCN neurons in freely moving *Vip-GABA$_A$R*$^{-/-}$ mice (Fig 3) [12,53,54]. Excitation light sources were a 470-nm LED for detecting calcium-dependent jGCaMP7s fluorescence signal (F470) and a 415-nm LED for calcium-independent isos-bestic fluorescence signal (F415). The duration of excitation lights is 50 ms, and the onsets of the excitation timing of LEDs were interleaved. The lights passed through excitation bandpass filters, dichroic mirrors, and then to the animal via fiber-optic patch cords (BBP(4)_400/440/900-0.37_1m_FCM-4xFCM_LAF, MFP_400/440/LWMJ-0.37_1m_FCM-ZF2.5_LAF, Doric Lenses) and the implanted optical fiber. Subsequently, both signals were detected using a CMOS camera through the optical fibers, dichroic mirrors, and emission bandpass filters. The recorded signals were acquired using Bonsai software, with a sampling rate of 10 Hz for each color. The excitation intensities of the 470-nm and 415-nm LED at the animal side's patch cord tip were from 60 μW to 110 μW. We recorded the same for 30 s every 10 min for 3 weeks to reduce photobleaching. The detected GCaMP signal was averaged within a 30-s session. Ratio (R) was defined as the ratio between F470 (Ca$^{2+}$-dependent signal) and F415 (Ca$^{2+}$-independent signal) (F470/F415) for calibration and reducing motion artifacts.

To detrend the gradual decrease of the signal during recording days, ±12 h average from the time (145 points) was calculated as 24-h reference ($F_{24h}$ for the single-color system, $R_{24h}$ for the dual-color system). The data were subsequently detrended by the subtraction of $F_{24h}$ or $R_{24h}$ (ΔF or ΔR). Then, the ΔF/$F_{24h}$ or ΔR/$R_{24h}$ value was calculated [12]. For the actogram and daily profile analyses of VIP-Ca$^{2+}$ rhythms, which were performed via ClockLab (Actimetrics), all ΔF or ΔR values were converted to positive values by subtracting the minimum value of ΔF or ΔR. Subsequently, these values were multiplied by 100 or 1,000 and rounded off. For VIP-Ca$^{2+}$ daily profiles, data during the last 4 (ΔF) or 5 (ΔR) days in the LD condition and last 5 (ΔF) or 7 (ΔR) days in the DD condition were used. To determine VIP-Ca$^{2+}$ onset and VIP-Ca$^{2+}$ offset, ΔF or ΔR was smoothened with a 11-point moving average before generating daily profiles. Then, the time points at which the signal crossed upward and downward thresholds corresponding to 20%, 50%, or 80% of the peak-to-trough amplitude

(maximum – minimum ΔF or ΔR) were defined as the onset and offset of high VIP-Ca$^{2+}$ activity, respectively, and the interval between VIP-Ca$^{2+}$ onset and offset was defined as high VIP-Ca$^{2+}$ duration. The resulting profiles were further normalized such that the minimum and maximum values were set to 0 and 1, respectively. CT12 was determined as the onset of locomotor activity, as described below. To quantify the speed of VIP-Ca$^{2+}$ increase and decline in the morning and evening, the upward and downward slopes were calculated by dividing the magnitude of VIP-Ca$^{2+}$ change (60% change: from 20% to 80% or from 80% to 20%) by the time (h) required for that change. To quantify the fluctuation of VIP-Ca$^{2+}$ signal, the CV was calculated based on the deviation of the unsmoothed daily profile from the corresponding smoothed daily profile. Recorded Ca$^{2+}$ rhythm patterns were consistent between single-color and dual-color fiber photometry systems, indicating that the overall conclusions were independent of the recording configuration.

During the fiber photometry recordings, the animal's locomotor activity (home-cage activity) was monitored using an infrared sensor (Supermex PAT.P and CompACT AMS Ver. 3, Muromachi Kikai) in 1-min bins, then 10-min bins were made by analysis. Data were analyzed using ClockLab (Actimetrics). A double-plotted actogram of locomotor activity was prepared and overlaid on that of the GCaMP. The onset and offset of locomotor activity were calculated from the daily locomotor activity profiles of the same days used for VIP-Ca$^{2+}$ profiles, regarding the median activity level as a threshold. The intervals between locomotor activity onset and offset were defined as locomotor activity time. The phase relationship between locomotor activity and VIP-Ca$^{2+}$ onset/offset was determined in individual mice by comparing the daily profiles of locomotor activity and smoothened VIP-Ca$^{2+}$ rhythm.

We confirmed the jGCaMP7s or gRNA (mCherry) expression and the position of the optical fiber by slicing the brains into 30 or 100 μm coronal sections using a cryostat (Leica). The sections were mounted on glass slides with a mounting medium (VECTASHIELD HardSet with DAPI, H-1500, Vector Laboratories or Dako Fluorescence Mounting Medium, Agilent Technologies) and observed via epifluorescence microscope (KEYENCE, BZ-9000E).

**Behavioral analyses**

Male and female *Vip-GABA$_A$R$^{-/-}$* (*Vip-ires-Cre; Rosa26-LSL-SpCas9-2A-EGFP* injected bilaterally with AAV-*U6-gGabrb1~3-EF1α-DIO-mCherry* mixture) and control (injected with control AAV) (Fig 2C) mice, aged 8–20 weeks, were housed individually in a cage placed in a light-tight chamber (light intensity was approximately 100 lux). Spontaneous locomotor activity (home-cage activity) was monitored by infrared motion sensors (O'Hara) in 1-min bins as described previously [12]. Actogram, activity profile, and χ$^2$ periodogram analyses were performed via ClockLab (Actimetrics). The free-running period and amplitude (Qp values) were calculated for the last 7 days in DD by periodogram. The onset and offset of locomotor activity and the activity time were calculated from the daily activity profile of the same 7 days in DD as described above.

**In vivo optogenetic stimulation with fiber photometry**

We used 10 *Avp-Cre; Vip-tTA* mice (*n*=6 for ChrimsonR, *n*=4 for control, both male and female), 4 *Avp-Vgat$^{-/-}$ × Vip-tTA* (*Avp-Cre; Vgat$^{flox/flox}$; Vip$^{wt/tTA}$*), and 6 *Avp-Vgat$^{+/-}$ × Vip-tTA* (*Avp-Cre; Vgat$^{wt/flox}$; Vip$^{wt/tTA}$*) mice. 8 *Avp-Cre; Vip-tTA* in this section are from a previously used cohort [12]. We injected 0.8–1.0 μL of the mixture of viruses (AAV-*TRE-FLEXoff-jGCaMP7s* with AAV-*CAG-Flex-ChrimsonR-mCherry* or AAV-*EF1α-DIO-hM3Dq-mCherry* and AAV-*TRE-jGCaMP7s*) into the right SCN (posterior: 0.5 mm, lateral: 0.25 mm, depth: 5.7 mm from the bregma) and then implanted an optical fiber (400 μm core, N.A. 0.39, 6 mm, Thorlabs) above the SCN (posterior: 0.2 mm, lateral: 0.2 mm, depth: 5.3 mm from the bregma) with dental cement. The mice were used for experiments more than 2 weeks after the surgery.

The dual-color fiber photometry system (FP3002, Neurophotometrics) was used to record the Ca$^{2+}$ signal of SCN neurons with optogenetic stimulation in freely moving mice, as described above in "In vivo fiber photometry" (Fig 4) [53,54]. The excitation intensities of the 470-nm and 415-nm LEDs at the tip of patch cord on the animal side ranged 70–130 μW. Additionally, a 635-nm red laser (inside the fiber photometry system FP3002) was transmitted through the same optical

fibers with an intensity of 2–3 mW. During the 360 s (6 min) recording in every 2 h, optical stimulation (635 nm, 50 ms pulse, 5 Hz, 120 s, 600 pulses) was applied in the middle of the recording. Throughout the experiment, the mice were housed in a custom-made acrylic cage surrounded by a sound-attenuating chamber and maintained in a 12-h LD cycle.

The recorded data were interleaved to eliminate artifacts caused by red laser stimulation, and half of it was discarded. Consequently, the final sampling rate for the jGCaMP7s fluorescence signals at the 470-nm light excitation (F470) and the 415-nm excitation (F415) was 5 Hz. Ratio (R) was defined as the ratio between F470 and F415 (F470/F415) for calibration and reducing motion artifacts. To extract circadian changes in baseline fluorescent ratio ($R_0$), $R_0$ was defined as the mean R-value during −30–0 s. Because recordings were obtained over 2 days, a 48-h reference ($R_{48h}$) was calculated as the average $R_0$ across the two recording days. The baseline signal ($R_0$) was detrended by subtracting $R_{48h}$ and detrended $R_0$ values were expressed as $\Delta R/R_{48h}$ (%). For optogenetic responses, $\Delta R$ was defined as the difference between the mean R value during the late phase of stimulation period (90–120 s, $R_1$) and $R_0$. The response magnitude was calculated as $\Delta R/R_0$ (%). After the recordings were completed, we confirmed the jGCaMP7s and ChrimsonR-mCherry expressions and the position of the optical fiber by histology.

## Slice electrophysiology

*Vip-GABA$_A$R$^{-/-}$* (made as described above) mice were compared to control (*Vip-ires-Cre; Rosa26-LSL-Cas9-2A-EGFP* with AAV-*EF1a-DIO-mCherry* injected, 1.0 µL bilateral of SCN) mice. Both male and female mice aged 8–20 weeks were used. Coronal brain slices (250 µm thick), including the SCN, were prepared with a linear-slicer (NLS-MT, Dosaka EM), as described previously [55]. Under an upright fluorescence microscope (Olympus, BX51WI), we visually identified VIP neurons with EGFP fluorescence in the ventral SCN and non-VIP neurons without the fluorescence in the dorsal SCN. The gRNA AAV-infected VIP neurons were identified by additional mCherry fluorescence. For recording of non-glutamatergic spontaneous postsynaptic currents (PSCs), the slices were continuously perfused with an artificial CSF (ACSF) with the following composition (in mM): 125 NaCl, 2.5 KCl, 2 CaCl$_2$, 1 MgSO$_4$, 26 NaHCO$_3$, 1.25 NaH$_2$PO$_4$, 10 d-glucose, equilibrated with 95% O$_2$ and 5% CO$_2$, kept at 31 ± 1 °C, and further mixed with 10 µM 6-cyano-7-nitroquinoxaline-2,3 dione disodium (CNQX) and 25 µM D-(-)-2-amino-5-phosphonopentanoic acid (D-AP5) to block fast glutamatergic transmission. Whole-cell voltage-clamp recordings were made at −60 mV with borosilicate glass electrodes (5–6 MΩ) filled with an internal solution containing the following components (in mM): 87 CsCl, 20 TEA-Cl, 10 HEPES, 10 EGTA, 0.5 CaCl$_2$, 4 MgCl$_2$, 2 QX314, 4 Na-ATP, 0.4 Na-GTP, and 10 phosphocreatine (pH 7.3, adjusted with CsOH). A combination of an EPC10/2 amplifier and Patchmaster software (HEKA) was used to control membrane voltage and data acquisition. Series resistance was compensated routinely by 80%. The nonglutamatergic PSCs completely disappeared in the presence of 10 µM gabazine (SR95531, Tocris), showing that they were mediated by GABA$_A$ receptors [25]. The GABAergic PSCs recorded for 1–3 min were analyzed using the MiniAnalysis program (Synaptosoft). The events were picked by an amplitude threshold of 5 pA and confirmed visually to have a typical PSC waveform.

## Ex vivo optogenetic stimulation and calcium imaging

We used 17 *Avp-Cre; Vip-tTA* mice (n = 6 (ZT22) or 5 (ZT14) for ChrimsonR, n = 6 (ZT22) for control) for VIP-Ca$^{2+}$ recording and 9 *Avp-Cre* mice for recording Ca$^{2+}$ in non-AVP neurons (aged 8–20 weeks, including both males and females). We injected 0.6 µL of the mixture of viruses (VIP-Ca$^{2+}$: AAV-*TRE-FLEXoff-jGCaMP7s* with AAV-*CAG-Flex-ChrimsonR-mCherry* or AAV-*EF1α-DIO-mCherry,* non-AVP-Ca$^{2+}$: AAV-*EF1α-rDIO-jGCaMP7s* with AAV-*CAG-Flex-ChrimsonR-mCherry*) into the SCN bilateral (posterior: 0.5 mm, lateral: 0.25 mm, depth: 5.7 mm from the bregma). Coronal brain slices (300 µm thick), including the SCN, were prepared as described in "Slice electrophygiology" For this experiment, mice were sacrificed at ZT19–20 in darkness under a red light or at ZT11–12 in light. The slices were placed in an imaging chamber and continuously perfused with the ACSF. Before the recordings started around projected ZT22 or ZT14, the slices were pre-incubated in the experimental environment for at least one hour, which was critical to observe VIP-Ca$^{2+}$ response

to the optogenetic stimulation. Under the upright fluorescence microscope (Olympus, BX51WI), we visually identified AVP neurons with ChrimsonR-mCherry or mCherry (control) fluorescence in the dorsomedial SCN and VIP neurons with jGCaMP7s fluorescence in the ventral SCN. We imaged jGCaMP7s fluorescence every 5 s through an optical filter set (an excitation filter 470–495 nm, a dichroic mirror 505 nm and an emission filter 510–550 nm) and a digital CMOS camera (Prime BSI Express, Teledyne Vision Solutions) with MetaFluor software.

After the jGCaMP7s signal had stabilized, we optogenetically stimulated SCN AVP neurons for 2 min (617 nm, 10 Hz, 40 ms duration) through an optic fiber (200 µm core, N.A. 0.39, 6 mm, ferrule 1.25 mm, FT200EMT-CANNULA; Thorlabs) positioned near the AVP neurons and connected to a high-powered LED under the control of Digital Stimulator (WPI DS8000B) and LED driver (THORLABS). Gabazine (10 µM) was applied to ACSF to verify $GABA_A R$ involvement. For VIP-$Ca^{2+}$, we measured the fluorescence values of the entire jGCaMP7s-expressing ROIs using MetaFluor software, detrended the traces by division, and calculated the change in fluorescence over baseline fluorescence ($\Delta F/F_0$%) as (Fi-$F_0$) * 100/ $F_0$. For time-course analyses, fluorescence signals were expressed as $\Delta F/F_0$ (%), where $F_0$ was defined according to the type of manipulation. For optogenetic stimulation analyses, $F_0$ was defined as the mean F value during −60 to −30 s relative to stimulation onset. For drug application analyses, $F_0$ was defined as the mean F value during −120 to −60 s to avoid overlap with subsequent comparison windows. $\Delta F$ was first calculated at each time point as the difference between the fluorescence signal (F) and the corresponding $F_0$. Then, the $\Delta F/F_0$ (%) value was calculated. To quantify response magnitude, $\Delta F/F_0$ values were then averaged within specified analysis time windows.

For non-AVP-$Ca^{2+}$, we selected unilateral SCN regions (150 × 260 pixels, 2.6 µm/pixel). Images were analyzed using MATLAB. The resolution of images was then adjusted to 20.8 µm/pixel. Fluorescence signals were analyzed in a two-step procedure consisting of exploratory (qualitative) pixel-wise analysis followed by ROI-based quantitative analysis. In the exploratory analysis, fluorescence signals were quantified on a pixel-by-pixel basis to generate $\Delta F/F_0$ (%) heatmaps, which were used to visualize the spatial distribution of $Ca^{2+}$ responses. For this purpose, $F_0$ was defined as the mean F value during −120–0 s. Based on these heatmaps, two representative regions exhibiting prominent responses (middle and ventral SCN) were selected for further quantitative analysis. For subsequent analyses, square regions of interest (ROIs; 4 × 4 pixels) were defined in the middle and ventral of the SCN regions. ROI-based fluorescence signals were then analyzed using the same analytical framework as described above. For these analyses, $F_0$ was defined as the mean F value during −60 to −30 s for both optogenetic stimulation and drug application experiments.

## Statistical analysis

All results are expressed as mean ± SEM. For comparisons of two groups, two-tailed Student *t* test or Welch's *t* test was performed. For comparisons of multiple groups with no difference of variance, two-way mixed-design ANOVA with time or lighting conditions as a within-subject (repeated-measures) factor and genotype as a between-subject (independent design) factor, followed by post-hoc two-tailed Student's or Welch's *t* test with Bonferroni correction were performed. For comparisons of multiple groups with difference of variance, nonparametric tests, Kruskal–Wallis test with post-hoc Dunn's test were performed. All *P* values less than 0.05 were considered as statistically significant. Only relevant information from the statistical analysis was indicated in the text and figures. For circular data, Watson–Williams test was performed with Circstat MATLAB Toolbox for Circular Statistics [56]

## Supporting information

**S1 Fig. Daily locomotor activity and VIP-$Ca^{2+}$ activity profiles of individual *Avp-Vgat*−/− mice in DD.** Data of 5 control and 8 *Avp-Vgat*$^{-/-}$ mice are shown. Daily VIP-$Ca^{2+}$ profiles (green) are overlaid with those of locomotor activity (black). VIP-$Ca^{2+}$ signals with detrending but without smoothening and normalization in the last 5 days in DD are shown here. Quantification of VIP-$Ca^{2+}$ in Figs 1 and S1 was performed using smoothened and normalized data. Black and gray circles indicate the timings of locomotor activity onset and offset, respectively. The black dashed line indicates the median

of locomotor activity, which was regarded as the threshold for determining the onset and offset. Values are mean±SEM of 5-day recordings. The large SEM in the locomotor activity rhythm indicates that the activity level at that time varied substantially from day to day.
(TIF)

**S2 Fig. Circadian rhythms of the behavior and VIP-Ca²⁺ in *Avp-Vgat−/−* mice.** **(A)** Trajectories of jGCaMP7s fluorescence from SCN VIP neurons in vivo in individual mice for 15 days (5 days in LD, 10 days in DD). Detrended and smoothened $\Delta F/F_0$ values are plotted. **(B)** Mean VIP-Ca²⁺ level at the onset (left) and offset (right) of locomotor activity (LMA) rhythm of control (blue) and *Avp-Vgat⁻ᐟ⁻* (red) mice in LD or in DD. Normalized and smoothened daily VIP-Ca²⁺ profiles were compared with normalized locomotor activity profiles in LD (LD2-5) and DD (DD8-12). **(C)** Mean coefficient of variation (CV, %) of VIP-Ca²⁺ activity during ZT2–9 or CT2–9, quantified as the temporal variability of the unsmoothed signal relative to the corresponding smoothed daily VIP-Ca²⁺ profile. **(D, E)** Mean amplitude (Qp values, D) and free-running period (E) of the locomotor activity rhythm in DD (last 5 days). **(F)** Mean onset (circle) and offset (square) times of locomotor activity (black) and VIP-Ca²⁺ (green) rhythms, calculated using thresholds set at 20%, 50%, or 80% of the peak-to-trough amplitude, in LD (left) or in DD (right). **(G)** Actograms shown separately for behavior and VIP-Ca²⁺ signals, replotted from the overlaid actogram in Fig 1C. Control (Blue, *Avp-Vgat⁺ᐟ⁻*, i.e., *Avp-Cre; Vgat^{wt/flox}*, n = 5); *Avp-Vgat⁻ᐟ⁻* (Red, *n* = 8). Values are mean±SEM. \*$P$<0.05; \*\*$P$<0.01; \*\*\*$P$<0.001; \*\*\*\*$P$<0.0001 by two-way mixed-design ANOVA post-hoc two-tailed Student *t* test with Bonferroni correction (B, C); two-tailed Student *t* test (D, E) or Watson–Williams test (F). The data underlying this figure can be found in S1 Data.
(TIF)

**S3 Fig. Daily locomotor activity profile of individual control mice in DD.** Data of 8 control mice (*Rosa26-CAG-LSL-SpCas9-2A-EGFP; Vip-ires-Cre* mice injected with AAV-*U6-gControl-EF1α-DIO-mCherry*) are shown. Representative coronal sections with mCherry expression are also presented to show the efficient and consistent AAV infection across individual mice. Black and gray circles indicate the timings of locomotor activity onset and offset, respectively. The black dashed line indicates the median of locomotor activity, which was regarded as the threshold for determining the onset and offset. Values are mean±SEM of 7-day recordings. The large SEM in the locomotor activity rhythm indicates that the activity level at that time varied substantially from day to day.
(TIF)

**S4 Fig. Daily locomotor activity profile of individual *Vip-GABA_AR−/−* mice in DD.** Data of 16 *Vip-GABA_AR⁻ᐟ⁻* mice are shown. Representative coronal sections with mCherry expression are also presented to show the efficient and consistent AAV infection across individual mice. Black and gray circles indicate the timings of locomotor activity onset and offset, respectively. The black dashed line indicates the median of locomotor activity, which was regarded as the threshold for determining the onset and offset. Values are mean±SEM of 7-day recordings. The large SEM in the locomotor activity rhythm indicates that the activity level at that time varied substantially from day to day.
(TIF)

**S5 Fig. Locomotor activity rhythms of *Vip-GABA_AR−/−* and *Avp-Vgat−/−; Vip-GABA_AR−/−* mice.** **(A, B)** Mean amplitude (Qp values, A) and free-running period (B) of the locomotor activity rhythm in DD (last 7 days). Blue, Control; red, *Vip-GABA_AR⁻ᐟ⁻*. Values are mean±SEM. *n* = 8 for control, *n* = 16 for *Vip-GABA_AR⁻ᐟ⁻* mice. \*\*$P$<0.01 by two-tailed Student *t* test. The data underlying this figure can be found in S1 Data.
(TIF)

**S6 Fig. Daily locomotor activity and VIP-Ca²⁺ activity profiles of individual *Vip-GABA_AR−/−* mice in DD.** Data of 9 control and 6 *Vip-GABA_AR⁻ᐟ⁻* mice are shown. Daily VIP-Ca²⁺ profiles (green) are overlaid with those of locomotor activity (black). VIP-Ca²⁺ signals with detrending but without smoothening and normalization in the last 7 days in DD are shown

here. Quantification of VIP-Ca$^{2+}$ in Figs 3 and S6 was performed using smoothened and normalized data. Black and gray circles indicate the timings of locomotor activity onset and offset, respectively. The black dashed line indicates the median of locomotor activity, which was regarded as the threshold for determining the onset and offset. Values are mean ± SEM of 7-day recordings. The large SEM in the locomotor activity rhythm indicates that the activity level at that time varied substantially from day to day.
(TIF)

**S7 Fig. Circadian rhythms of the behavior and VIP-Ca$^{2+}$ in *Vip-GABA$_A$R−/−* mice. (A)** Trajectories of jGCaMP7s fluorescence from SCN VIP neurons in vivo in individual mice for 17 days (5 days in LD, 12 days in DD). Detrended and smoothened $\Delta F/F_0$ values are plotted. **(B)** Mean VIP-Ca$^{2+}$ level at the onset (left) and offset (right) of locomotor activity (LMA) rhythm of control (blue) and *Avp-Vgat$^{-/-}$* (red) mice in LD or in DD. Normalized and smoothened daily VIP-Ca$^{2+}$ profiles were compared with normalized locomotor activity profiles in LD (LD3-7) or in DD (DD8-14). **(C)** Mean coefficient of variation (CV, %) of VIP-Ca$^{2+}$ activity during ZT2–9 or CT2–9, quantified as the temporal variability of the unsmoothed signal relative to the corresponding smoothed daily VIP-Ca$^{2+}$ profile. **(D, E)** Mean amplitude (Qp values, D) and free-running period (E) of the locomotor activity rhythm in DD (last 7 days). **(F)** Average locomotor activity counts per 6-hour interval in LD (left) or DD (right). **(G)** Mean onset (circle) and offset (square) times of locomotor activity (black) and VIP-Ca$^{2+}$ (green) rhythms, calculated using the thresholds set at 20%, 50%, or 80% of the peak-to-trough amplitude) rhythms in LD (left) or in DD (right). **(H)** Actograms shown separately for behavior and VIP-Ca$^{2+}$ signals, replotted from the overlaid actogram in Fig 3C. Control (Blue, *n* = 9); *Vip-GABA$_A$R$^{-/-}$* (Red, *n* = 6). Values are mean ± SEM. *$P$ < 0.05; **$P$ < 0.01; ***$P$ < 0.001 by two-way mixed-design ANOVA post-hoc two-tailed Student *t* test with Bonferroni correction (B, C, F); two-tailed Student *t* test (D, E) or Watson–Williams test (G). The data underlying this figure can be found in S1 Data.
(TIF)

**S8 Fig. Optogenetic activation of AVP neurons increases VIP-Ca$^{2+}$ in vivo during the night. (A)** VIP-Ca$^{2+}$ $R_0$ (without detrend) at 12 time points of two consecutive days. $R_0$ (the baseline ratio) is the mean R-value of the late phase of pre-stimulation period (−30 to 0 s). Blue, control (*Avp-Vgat$^{+/-}$*, *n* = 6 mice); red, *Avp-Vgat$^{-/-}$*, *n* = 4 mice). White, day; dark gray, night. **(B)** Population mean of $R_0$ in (A). VIP-Ca$^{2+}$ $R_0$ over two days was averaged by time of day to generate a 24-h profile per mouse. Error bars are SEM. Two-way mixed-design ANOVA with ZT as a within-subject (repeated-measures) factor and genotype as a between-subject factor. **(C)** Comparison of the $\Delta R/R_0$% in *Avp-Cre; Vip-tTA* mice. The baseline ratio ($R_0$) is the mean R-value of pre-stimulation period (−30 to 0 s). $\Delta R$ is the difference between the mean R-value of the late phase during the stimulation period ($R_1$, 90–120 s) and $R_0$. Optogenetic stimulation of SCN AVP neurons increases VIP-Ca$^{2+}$ during the night in freely moving mice. *n* = 6 for ChrimsonR, *n* = 4 for control. ***$P$ < 0.001 by two-tailed Welch's *t* test. Four out of 6 in ChrimsonR and all control animals are from a previously used cohort [12]. **(D)** Representative traces of the jGCaMP7s signal of SCN VIP neurons upon optogenetic stimulation of AVP neurons at various timing in vivo (top, ChrimsonR; bottom, mCherry-Control). Red shading indicates the timing of optical stimulation (635 nm, 50 ms pulse, 5 Hz, 120 s). The data underlying this figure can be found in S1 Data.
(TIF)

**S9 Fig. Response of VIP-Ca$^{2+}$ to the optogenetic activation of AVP neurons in SCN slices. (A)** A representative jGCaMP7s trace of VIP neurons in an SCN slice with optogenetic stimulation of AVP neurons via ChrimsonR around ZT22, first without and then with gabazine application. Red and green shadings indicate the optical stimulation and gabazine application periods, respectively. **(B** and **C)** Traces of the jGCaMP7s signals from VIP neurons in individual SCN slices aligned to optogenetic stimulation of AVP neurons at ZT22 without (B) or with (C, red lines) gabazine application. Red shading indicates the timing of optical stimulation (617 nm, 40 ms pulse, 10 Hz, 120 s). *n* = 6 for control (mCherry) mice, *n* = 6 for ChrimsonR mice. The ChrimsonR traces in (C, blue) and (B, blue) are identical. **(D)** Averaged trace of jGCaMP7s signal in VIP neurons during bath application of gabazine (GABA$_A$R antagonist, 10 μM). The green shading

indicates the period of gabazine application. (**E**) Baseline-corrected mean jGCaMP7s signal before (−60 to 0 s in F) and during the last 60 s of gabazine application (180–240 s in F). Baseline ($F_0$) is defined as the mean fluorescence signal during $-120 - -60$ s. (**F**) Averaged jGCaMP7s traces from SCN VIP neurons aligned to optogenetic stimulation of AVP neurons without (ChrimsonR, blue) or with (ChrimsonR + Gabazine, red) gabazine application at ZT14. Red shading indicates the timing of optical stimulation (617 nm, 40 ms pulse, 10 Hz, 120 s). (**G**) Mean jGCaMP7s signal before (−30 to 0 s in D) and during the last 30 s of optogenetic stimulation (90 − 120 s in D) in the absence (left) or presence (right) of gabazine. Baseline ($F_0$) is defined as the mean fluorescence signal during −60 to −30 s. Values are mean ± SEM. $n = 5$ for ChrimsonR mice in ZT14. *$P < 0.05$ by two-tailed Paired $t$ test. The data underlying this figure can be found in S1 Data. (TIF)

**S10 Fig. Response of non-AVP cells to the optogenetic activation of AVP neurons in coronal SCN slices of individual mice. (A)** Left: jGCaMP7s expression in non-AVP cells of coronal SCN slices prepared from 9 *Avp-Cre* mice injected with AAV-*EF1α-rDIO-jGCaMP7s* and AAV-*CAG-Flex-ChrimsonR-mCherry*. White outlines indicate the regions considered as the SCN. Right: Pixel-level heat maps (20.8 μm/pixel) showing the jGCaMP7s fluorescence from non-AVP cells in response to optogenetic stimulation of AVP neurons at ZT22. Blue and red squares on the maps indicate regions (4 × 4 pixels) considered as the middle and ventral regions, respectively. Data from slice #2 are also used in Fig 6C. F value is a.u (arbitrary unit). (**B**) Traces of the jGCaMP7s signals from non-AVP cells in the middle (left) and ventral (right) regions in individual SCN slices in response to the optogenetic stimulation of AVP neurons. The data underlying this figure can be found in S1 Data. (TIF)

**S1 Data. Underlying numerical data.** Excel spreadsheets containing the individual subject-level data used to generate Figs 1–6 and S2, S5, S7–S10. (XLSX)

## Acknowledgments

We thank H. Okamoto for the *Avp-Cre* mouse; S. Horike and T. Daikoku for the *Vip-tTA* mouse; Z. J. Huang for the *Vip-ires-Cre* mouse; F. Zhang for the *Rosa26-LSL-SpCas9-2A-EGFP* mouse; B.B. Lowell for the *Vgat^flox* mouse; Penn Vector Core for *pAAV2-rh10*; D. Kim and GENIE Project for *pGP-AAV-CAG-FLEX-jGCaMP7s-WPRE*; A. Ting for *pAAV-TRE-ChrimsonR-mCherry*; A. Ventura for *pX333*; B. Roth for *pAAV-EF1a-DIO-mCherry* and *pAAV-EF1α-DIO-hM3Dq-mCherry*; and *K. Deisseroth* for *pAAV-DIO-hChR2(H134R)-EYFP-WPRE-pA*; We thank all lab members, including M. Kawabata and Y. Nishiwaki.

## Author contributions

**Conceptualization:** Yubo Peng, Yusuke Tsuno, Michihiro Mieda.

**Data curation:** Yubo Peng, Yusuke Tsuno, Takashi Maejima, Mohan Wang, Jaehun Jung.

**Formal analysis:** Yubo Peng, Yusuke Tsuno, Takashi Maejima, Mohan Wang, Jaehun Jung.

**Funding acquisition:** Yubo Peng, Yusuke Tsuno, Takashi Maejima, Ayako Matsui, Michihiro Mieda.

**Investigation:** Yubo Peng, Yusuke Tsuno, Takashi Maejima, Mohan Wang, Jaehun Jung, Michihiro Mieda.

**Methodology:** Yubo Peng, Yusuke Tsuno, Takashi Maejima, Michihiro Mieda.

**Project administration:** Yubo Peng, Yusuke Tsuno, Takashi Maejima, Michihiro Mieda.

**Resources:** Jaehun Jung, Ayako Matsui, Michihiro Mieda.

**Supervision:** Michihiro Mieda.

**Validation:** Yubo Peng, Yusuke Tsuno, Takashi Maejima.

**Visualisation:** Yubo Peng, Yusuke Tsuno, Takashi Maejima.

**Writing – original draft:** Yubo Peng, Yusuke Tsuno, Takashi Maejima, Michihiro Mieda.

**Writing – review & editing:** Yusuke Tsuno, Michihiro Mieda.

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
