## [Editor Report · Decision Letter 0]

9 Feb 2026

Dear Dr Mieda,

Thank you for submitting your manuscript entitled "GABAergic network from AVP neurons to VIP neurons in the suprachiasmatic nucleus sets the activity/rest time of the circadian behavior rhythm", which has been revised in response to a previous round of review, for consideration as a Research Article by PLOS Biology.

Your revised manuscript has now been evaluated by the PLOS Biology editorial staff and I am writing to let you know that we would like to send your submission back to the original reviewers for another look.

Once your full submission is complete, your paper will undergo a series of checks in preparation for peer review. After your manuscript has passed the checks it will be sent out for review. To provide the metadata for your submission, please Login to Editorial Manager (https://www.editorialmanager.com/pbiology) within two working days, i.e. by Feb 11 2026 11:59PM.

Kind regards,

Luke

Lucas Smith, Ph.D.

Senior Editor

PLOS Biology

lsmith@plos.org

---

## [Editor Report · Decision Letter 1]

19 Feb 2026

Dear Dr Mieda,

Thank you again for submitting a revised version of your manuscript "GABAergic network from AVP neurons to VIP neurons in the suprachiasmatic nucleus sets the activity/rest time of the circadian behavior rhythm" for consideration as a Research Article at PLOS Biology. Since my last email, I was actually able to discuss your revision with the Academic Editor (who had also provided reviewer comments as reviewer 3) and s/he is fully satisfied by the changes made in response to the last round of review.

Based on our Academic Editor's assessment of your revision, we are likely to accept this manuscript for publication. However, before we can do so we need you to address a few last minor editorial requests in a last revision that should not take very long. These are detailed below.

**IMPORTANT - Please address the following editorial requests:

1) TITLE: We would like to propose a tweak to the title that we think improves its clarity. If you agree, we suggest you change the title to:

"A GABAergic connection from AVP- to VIP-neurons in the suprachiasmatic nucleus sets the timing of circadian behavior rhythms"

or maybe

"A GABAergic network from AVP- to VIP-neurons in the suprachiasmatic nucleus sets the timing of circadian behavior rhythms"

2) DATA: Thank you very much for providing the underlying data for your study as supplemental file S1_data. Please add a sentence to each figure legend, pointing readers to this file. (Ex you can say something like 'The data underlying this figure can be found in S1_data')

3) CODE: Per journal policy, if you have generated any custom code during the course of this investigation, please make it available without restrictions. Please ensure that the code is sufficiently well documented and reusable, and that your Data Statement in the Editorial Manager submission system accurately describes where your code can be found. More information on our Code Policy, what and how to share can be found here: https://journals.plos.org/plosbiology/s/code-availability

We expect to receive your revised manuscript within two weeks.

*Published Peer Review History*

*Press*

Sincerely,

Luke

Lucas Smith, Ph.D.

Senior Editor

lsmith@plos.org

PLOS Biology

---

## [Editor Report · Decision Letter 2]

27 Feb 2026

Dear Dr Mieda,

Thank you for the submission of your revised Research Article "A GABAergic network from AVP- to VIP-neurons in the suprachiasmatic nucleus sets the timing of circadian behavior rhythms" for publication in PLOS Biology. On behalf of my colleagues and the Academic Editor, Horacio de la Iglesia, I am pleased to say that we can in principle accept your manuscript for publication, provided you address any remaining formatting and reporting issues. These will be detailed in an email you should receive within 2-3 business days from our colleagues in the journal operations team; no action is required from you until then. Please note that we will not be able to formally accept your manuscript and schedule it for publication until you have completed any requested changes.

While you attend to those requests to come, could you please also add a sentence to each figure legend of the figures in the supplementary information where the source data can be found?

PRESS

Sincerely,

Christian

Christian Schnell, PhD

Senior Editor and Team Manager, PLOS Biology

cschnell@plos.org

on behalf of

Lucas Smith, Ph.D.

Senior Editor

PLOS Biology

lsmith@plos.org